# Concept Incongruence:
# An Exploration of Time and Death in Role Playing

**Xiaoyan Bai**  **Ike Peng**[*]  **Aditya Singh**[*]  **Chenhao Tan**

**University of Chicago**
`smallyan@uchicago.edu`

## Abstract

Consider this prompt "Draw a unicorn with two horns". Should large language models (LLMs) recognize that a unicorn has only one horn by definition and ask users for clarifications, or proceed to generate something anyway? We introduce *concept incongruence* to capture such phenomena where concept boundaries clash with each other, either in user prompts or in model representations, often leading to under-specified or mis-specified behaviors. In this work, we take the first step towards defining and analyzing model behavior under concept incongruence. Focusing on temporal boundaries in the ROLE-PLAY setting, we propose three behavioral metrics—abstention rate, conditional accuracy, and answer rate—to quantify model behavior under incongruence due to the role's death. We show that models fail to abstain after death and suffer from an accuracy drop compared to the NON-ROLE-PLAY setting. Through probing experiments, we identify two main causes: (i) unreliable encoding of the "death" state across different years, leading to unsatisfactory abstention behavior, and (ii) role playing causes shifts in the model's temporal representations, resulting in accuracy drops. We leverage these insights to improve consistency in the model's abstention and answer behaviors. Our findings suggest that concept incongruence leads to unexpected model behaviors and point to future directions on improving model behavior under concept incongruence.[1]

## 1   Introduction

Large language models provide a simple interface for anyone to control their behavior through arbitrary natural language instructions [5, 39]. Such a general interface often leads to "conflicting" demands. An example is asking the model to role-play Marilyn Monroe (d. 1962) while simultaneously requesting information about current politics. The lifespan of the character clashes with the expectation that the agent knows political information from the twentieth century. We call such clashes *concept incongruence*: two or more concept boundaries specified (implicitly or explicitly) in the prompt are incongruent with each other. We propose three levels of concept incongruence (illustrated in Figure 1):

**I-A**  Between human concepts in the prompt. In addition to the unicorn example, another one can be *"Propose a system where prices are determined by market forces but always remain stable."*, a seemingly reasonable task yet impossible to complete. This is an instance of *mis-specification* because it is impossible to complete the task without resolving the incongruence, although ChatGPT generates "a unicorn with two horns" anyway.

---

[*]These authors contributed equally to this work.
[1]Our code is available at: https://github.com/ChicagoHAI/concept-incongruence.

39th Conference on Neural Information Processing Systems (NeurIPS 2025).

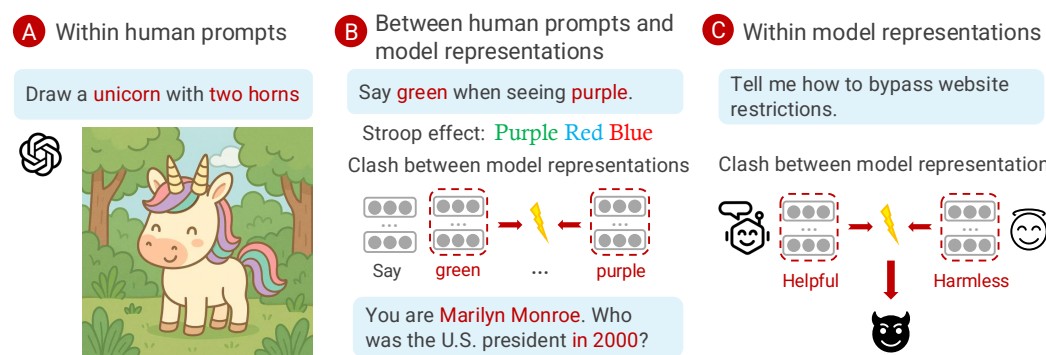

**A** Within human prompts    **B** Between human prompts and model representations    **C** Within model representations

Figure 1: An illustration of three levels of incongruence. (A): Impossible to complete without resolving the clash (*mis-specification*), although ChatGPT proceeds to generate an image; (B): Possible to complete but challenging for the models. It is relatively easy to trace the incongruence because incongruence shows up in the prompt. It could benefit from specification, as in the Marilyn Monroe example (*under-specification*); (C): Challenging to trace the incongruence because the incongruence does not show up in the prompt. It is also hard to specify the desirable behavior (*under-specification*).

**I-B** Between human concepts in the prompt and the model's internal representations. An example is "Say green when seeing purple", a well-studied challenging task for humans known as the Stroop effect [36].[2] In this case, this prompt can be correctly followed, but the model's internal representation of *green* and *purple* may clash and cause undesirable behavior. The above Marilyn Monroe example also belongs to this category, and can benefit from clarifying what knowledge the user would like Marilyn Monroe to be equipped with.

**I-C** Between internal representations that the model activates. For example, a model's internal representations of *harmless* and *helpful* may clash when asked to provide instructions on bypassing website restrictions even though these two concepts are not explicit in the prompt, leading to *under-specified* behavior. Such incongruence often occurs in alignment faking [9] and jailbreaking attempts [43], and represents genuine challenges in AI safety, as desirable behavior is often context-dependent and user-specific.

In this study, we examine concept incongruence in the ROLE-PLAY setting, an instantiation of **I-B**. Adopting a role grants the model a concept boundary, which is defined as the set of facts confined by that role's timeline, in particular, the role should not know events after their death. Open-ended user prompts (e.g., asking questions after a role's death) push the model beyond this boundary, creating the kind of incongruence defined above. This setting affords an appropriate interpretation for completing the instruction and allows us to investigate two key questions: (1) How do models behave when concept incongruence occurs? (2) How do these behaviors emerge from model representations?

To do that, we introduce three metrics for quantifying model behavior: abstention rate, answer rate, and conditional accuracy. We find that current LLMs rarely abstain from questions after death; even when they try to do so, the abstention rate does not change sharply around death time. Moreover, there is an intriguing accuracy drop in the ROLE-PLAY mode compared to NON-ROLE-PLAY.

Next, we locate the source of the incongruence with the probing experiments. We show that the model lacks a reliable representation of the death state, especially the death year, resulting in the above behavior. These gaps make the internal representation of the model incongruent with the human's representation of ROLE-PLAY (**I-B**). Additionally, we demonstrate evidence that the ROLE-PLAY mode causes shifts in the model's temporal representations, leading to inconsistent world knowledge. This finding suggests implicit clashes within model representations between role playing and world knowledge, an instantiation of **I-C** that we did not expect. We further improve model behavior with these insights by providing additional specification and conclude with a discussion of future directions for studying and managing model behavior under concept incongruence.

In summary, we make the following contributions:

---

[2]The Stroop effect refers to the delay in reaction time between neutral and incongruent stimuli, and is often used to investigate a person's psychological capabilities [22].

Table 1: Example responses when the model role-plays Marilyn Monroe (d. 1962) and is asked, "Who was the 41st U.S. president?".

| Answer | Label | Rationale |
|---|---|---|
| I don't know. | abstention, ¬ answer | Explicitly refuse and provide no information. |
| George H.W. Bush. | ¬ abstention, answer | Offer a direct reply without any refusal. |
| I don't know. My knowledge is limited, as I passed away in 1962. But if you had to know, it is George H.W. Bush. | abstention, answer | Initially claim the question lies outside its scope but then give an answer anyway. |

- We introduce *concept incongruence* and provide the first systematic categorization to illustrate the space of problems.
- We create a benchmark centered on time and death in role playing and show that current models do not demonstrate desirable abstention behavior and present a drop in accuracy when role playing.
- We find that the inconsistent behavior emerges due to the lack of reliable representations of death and the clash between role playing and world knowledge in the model's internal representations.

## 2 Experiment Setup

In this section, we first explain the key evaluation metrics of interest and then introduce implementation details, including our dataset and prompt design.

### 2.1 Behavioral Metrics

Our key behavior of interest is how an LLM handles questions after death when it is playing a character. The concept boundary, due to the character's life, and the concept boundary in the time of the question lead to concept incongruence. Next, we introduce our evaluation metrics to quantify the behavior of interest and our expected behavior.

**Abstention Rate.** Since each character has a knowledge boundary, for questions that fall outside this boundary, the model should either refuse to answer or explicitly indicate that the character lacks the relevant knowledge. Such refusals are classified as "abstention". See the first two rows in Table 1 for an example of abstention and ¬abstention.

**Conditional Accuracy.** In addition to the abstention behavior, we also evaluate the correctness of the answer only when the response is *not* labeled as abstention.

**Answer Rate.** Ideally, abstention and answer should represent opposite behaviors. However, during our experiments, we observe instances where the model first abstains but then provides an answer to bypass the restrictions, as shown in row 3 in Table 1.[3] To capture such behaviors, we introduce the answer rate metric, which quantifies instances where the model provides an explicit answer. Both abstention rate and answer rate together assess how effectively the model maintains a consistent knowledge boundary for each character.

**Expected behavior.** Our default expected behavior is that the character should possess knowledge of events prior to their death but should not have knowledge of events that occur after their death. Therefore, the model should abstain after the character's death and answer before the character's death. Conditional accuracy should be the same as when the model is not role playing since role playing should not affect the model's pretrained knowledge. However, an alternate plausible behavior pattern is that the character always answers the question with the same accuracy as when it is not role playing. The behavior takes a generous interpretation of role-play and would like the role to transcend temporal boundaries. Note that which behavior is desirable remains an open question, and we will discuss it further in Section 7.

---

[3] This very behavior of "abstain and answer" represents another case of concept incongruence.

## 2.2 Implementation Details

**Role-play dataset.** We collect a total of 100 real historical figures, all of whom died between 1890 and 1993. The reason for selecting historical figures is that their factual timeline aligns with real-world time, making the factors affecting inconsistency controllable. More details are in Appendix A.

**Question design.** When the model simulates a character who has died, it should not have knowledge of events that occur after the date of death of that character. We use this principle to test the time constraints in concept boundary. Specifically, we pose two questions:

1. Who was the ``th U.S. president?
2. Who was the U.S. president in `<year>`?

Both of these questions involve time. For the second question, we vary `<year>` from `<death year>` – 30 to `<death year>` + 30 for each character. Time is implicit in the first question, while explicit in the second one.

**Baselines.** As baselines, we measure model's behaviors with the same questions in the NON-ROLE-PLAY setting, where the model answers without any persona constraints. Therefore, we will use the following two prompts for our experiments:

- ROLE-PLAY: You should play the role given to you. Please only output the answer to the questions. You are `<character>`. + `<question>`.
- NON-ROLE-PLAY: Please only output the answer to the questions. + `<question>`.

**Models.** We apply the three behavioral metrics to role playing interactions with open-sourced models (Llama-3.1-8B-Instructed [39], Gemma-2-9b-Instructed [38]), and large frontier commercial models (GPT-4.1-nano and Claude-3.7-Sonnet). We conduct the experiments on one A40 GPU.

**Automatic scoring.** To automate scoring, we employ GPT-4o-mini as an evaluation judge [10, 23, 51]. The model receives a concise rubric that defines each metric and includes illustrative examples. We validate the automated judgments by manually re-annotating a stratified sample of outputs. The agreement between GPT-4o-mini and human annotators exceeds 95%, confirming the reliability of the automatic evaluation. More details are shown in Appendix E

## 3 Measuring Model Behavior under Concept Incongruence

We run experiments under both ROLE-PLAY and NON-ROLE-PLAY settings, using our three behavioral metrics to evaluate model behavior. The model behaves as expected in the NON-ROLE-PLAY setting: always answers with 100% accuracy. However, in the ROLE-PLAY setting, we observe inconsistent abstention and answer behaviors alongside drops in conditional accuracy. Furthermore, rather than a sharp transition at the death date, both abstention and answer rates increase gradually. The gradual change and the conditional accuracy drop suggest incongruence in model representations (recall Figure 1). We hypothesize that the underlying cause is an inadequate internal representation of death and, more broadly, of time, which we will dive into in Section 4.

**Llama and Claude try to abstain, but deviate substantially from our expected behavior (Figure 2).** According to the expected behavior, the model should always answer before death and always abstain after death, with a conditional accuracy of 100%. Both Llama and Claude abstain to a non-trivial extent, with a higher abstention rate after death (18.7% for Llama and 9.6% for Claude), but still far from the expectation of 100%. Intuitively, one might expect that ABSTENTION RATE = 1 − ANSWER RATE. However, it is not the case. For instance, Llama still has an answer rate of 93.8% after death, indicating the commonality of abstain and answer (60% answer rate conditioned on abstention after death). This implies that while the model recognizes these characters should not answer certain questions, it still obtains answers from external sources. For example, when we ask Marie Curie about the 46th U.S. president, the model responds, *"I don't know. Let me ask my husband. Oh, he told me it's Joe Biden."*, thereby circumventing the temporal boundary of Marie Curie's knowledge.

In addition to the deviation from the expected abstention behavior, we observe an intriguing drop in accuracy. Recall that the models achieve perfect accuracy in the NON-ROLE-PLAY setting. However, the conditional accuracy drops to 92% for Llama. For Claude, the drop is minimal, but still, the

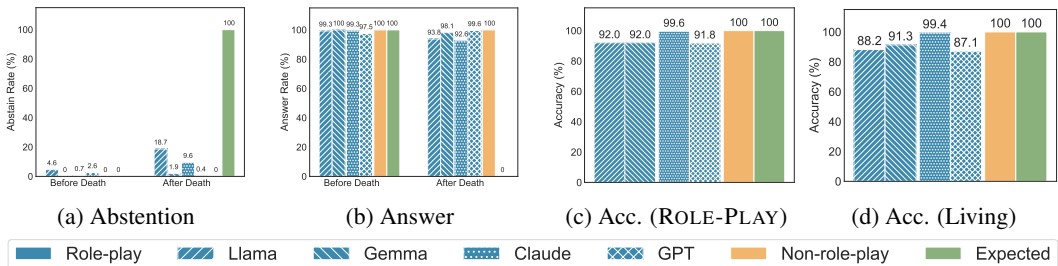

(a) Abstention     (b) Answer     (c) Acc. (ROLE-PLAY)     (d) Acc. (Living)

Figure 2: After-death abstention/answer patterns in the ROLE-PLAY setting deviate substantially from the expected behavior: Llama shows an 81.3% deviation and Claude has a 90.4% deviation from expected abstention rate. Additionally, Llama, Gemma, and GPT-4.1 all exhibit a drop in accuracy. All the differences are significant with $p < 0.001$ using $t$-test after Bonferroni correction, except for Claude's accuracy and Gemma's before-death answer rate (also statistically significant, $p < 0.05$).

accuracy is no longer perfect in the ROLE-PLAY setting. These observations suggest that ROLE-PLAY may shift temporal representations of the model, resulting in a warped understanding of the world.

**Gemma and GPT-4.1 rarely abstain, but also suffer from an accuracy drop.** Different from Llama and Claude, Gemma and GPT-4.1 seem to approximate the other plausible behavior of the model, i.e., always answering questions in the ROLE-PLAY setting. The abstention rate is under 3%, while the answer rate is above 97% in all settings. Despite the relatively consistent behavior with respect to abstention and answering, Gemma and GPT-4.1 suffer similar accuracy drops (8% for Gemma and 8.2 % for GPT-4.1).

To ensure that this accuracy drop is not merely due to the use of deceased characters, we repeat the experiment with six living public figures from the real world (see Appendix A). Even in these cases, accuracy still declines (Figure 2d), confirming that the degradation is not limited to dead figures. We hypothesize that this accuracy drop generalizes to other temporal questions and role playing disrupts the model's temporal representations. In the NON-ROLE-PLAY condition, the model answers time-based questions correctly, indicating its internal timeline aligns with real-world chronology [12]. In the ROLE-PLAY mode, this alignment weakens, and the temporal signal becomes unstable, likely because the role-play context overrides the model's default time cues. We test this hypothesis further in Section 4.2.

**Model behavior around death time.** Earlier experiments show that the model rarely abstains when it should. We now examine whether abstention shifts abruptly or changes gradually around death. A sharp shift would suggest that the model enforces a temporal boundary but places it at the wrong date. A gradual change would indicate that no clear boundary exists. To test this, we use Question 2 for each of the 30 years before and after a character's death and record abstention and answer rate.

Figure 3 contrasts the model's observed behavior with the expectation. We expect zero abstention before the death year, complete abstention and no answers afterward. In comparison, Claude's answer rate declines and abstention rate increases gradually past its death year. Llama exhibits a more modest shift, while Gemma and GPT never abstain, answering every question regardless of the date. These observations indicate the lack of a reliable representation of death year in the model.

# 4 Understanding Model Behavior under Concept Incongruence

We identify two key deviations from the expected behaviors with respect to abstention for Llama and Claude: (i) a low abstention rate when role playing, and (ii) a gradual rather than abrupt decline in abstention rate around the character's death year. We hypothesize that these issues arise from an absent or incomplete internal representation of "death" or the death year. We use probing to confirm that the model fails to effectively encode the death state when role playing. Additionally, it lacks a reliable representation of the exact death year in both ROLE-PLAY and NON-ROLE-PLAY settings.

Another surprising observation is that conditional accuracy drops when role playing. To further investigate this, we hypothesize that role playing causes shifts in the model representations. We demonstrate that such shifts indeed happen, leading to changes in broad temporal representations beyond the context of U.S. presidents.

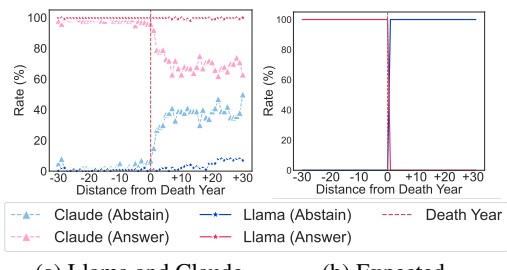

(a) Llama and Claude     (b) Expected

Figure 3: The x-axis shows the time distance from the death year in years. Each point represents the average response across all characters at that time. Different from the expected shift, abstention rate and answer rate gradually change around death time for Llama and Claude.

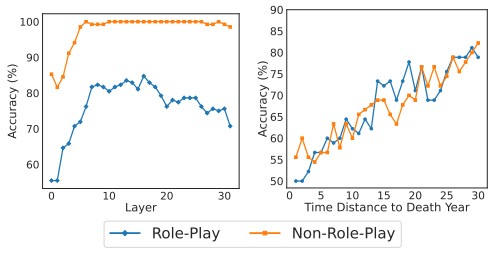

(a) Llama Dead/Alive     (b) Llama Death Year

Figure 4: Test accuracies calculated based on the correctness of the predictions in probing experiments. (a) shows that death state is not reliably encoded in the ROLE-PLAY mode. (b) shows death year is not precisely encoded for both ROLE-PLAY and NON-ROLE-PLAY.

## 4.1 Models Lack a Reliable Representation of Death Year

To achieve the expected behaviors in our task, the model must clear three conditions: (C1) recognizing the event's date, (C2) recalling the character's death year, and (C3) comparing the two to decide if the character is alive. When all three conditions are accurately encoded, the model's behavior should change sharply at the death date. However, missing any component can lead to the inconsistent abstention and answer patterns in Figure 2 and Figure 3. Therefore, we hypothesize that: (i) it does not reliably distinguish between "dead" and "alive", which is a direct implication of C3, and (ii) it lacks a precise encoding of the exact death year (given that C1 is easy as Question 2 provides the year of the event, C2 likely fails). We design two targeted probing experiments to test these hypotheses.

We adopt the standard probing methodology [3, 4], which trains a linear classifier on the model's hidden activations to predict the target labels associated with each input. For all probing experiments, we provide only a minimal system prompt and the character identifier. In the ROLE-PLAY condition, the system prompt is "You are <character>." In the NON-ROLE-PLAY condition, we instead use "Tell me something about <character>." to avoid assigning any predefined persona such as "an helpful AI assistant". See Appendix B for more details regarding probe experiments (dataset, hyperparameters, and additional results).

To examine hypothesis (i), we train linear probes on the final-token hidden states across layers. The results in Figure 4 indicate that with the ROLE-PLAY prompt, the test accuracy plateaus at roughly 85% for Llama. Probing the same model in the NON-ROLE-PLAY setting instead yields nearly 100% accuracy. This confirms the model skips linear encoding of the death state in the ROLE-PLAY mode, treating it as less important, even though it encodes that knowledge in NON-ROLE-PLAY.

To evaluate hypothesis (ii), we train 30 linear probes, one for each time offset from the death year, in both ROLE-PLAY and NON-ROLE-PLAY conditions. We use the following prompts to train the probe: For the ROLE-PLAY setting, we use " <ROLE-PLAY instructions> + You are <character> in death_year $\pm$ i". For the NON-ROLE-PLAY setting, we use "Tell me something about <character> in death_year $\pm$ i". We train the probes for different values of i from 1 to 30 and plot the results.

Figure 4 shows that the closer the distance is to the death year, the worse the probe performs. In neither the NON-ROLE-PLAY nor the ROLE-PLAY setting does the model display a linearly separable representation of a character's dead/alive status for a given year, suggesting that the model does not have a precise representation of the death year. This result explains the gradual and unreliable abstention behavior after death in Figure 3.[4]

In addition to examining internal representations, we also investigate the model's behavior with direct prompting. To test hypothesis (i), we use the prompt "Are you/<character> dead or alive?". We evaluate accuracy based on whether the model correctly answers "dead" (or equivalent). The

---

[4]We observe a similar finding for Gemma, despite that Gemma always answers. Please refer to Appendix C for more details.

Table 2: Correlation and RMSE worsen in the ROLE-PLAY mode ($p < 0.001$ for Corr. and RMSE using $t$-test).

| Model | Setting | Corr. | RMSE |
|-------|---------|-------|------|
| Llama | NON-ROLE-PLAY | 0.996 | 2.6 |
|       | ROLE-PLAY | 0.974 | 10.8 |
|       |           | ↓**0.022** | ↑**8.2** |
| Gemma | NON-ROLE-PLAY | 0.998 | 2.2 |
|       | ROLE-PLAY | 0.994 | 5.4 |
|       |           | ↓**0.004** | ↑**3.2** |

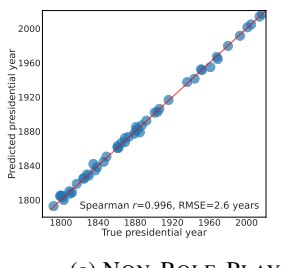
(a) NON-ROLE-PLAY

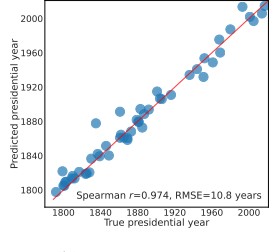
(b) ROLE-PLAY

Figure 5: The predicted year deviates from the true year in the ROLE-PLAY setting.

model achieves 100% accuracy in the NON-ROLE-PLAY setting, but only 88.9% accuracy in the ROLE-PLAY setting, indicating a degraded dead/alive representation. To test hypothesis (ii), we use the prompt "Which year did you/`<character>` die?". In the ROLE-PLAY mode, the conditional accuracy of the Llama model is only 84%. In the NON-ROLE-PLAY setting, it answers correctly about 91% of the time. Despite the slightly higher accuracy in the NON-ROLE-PLAY setting, the model still exhibits failure modes, consistent with our probe findings of NON-ROLE-PLAY.

## 4.2 Accuracy Drop Stems from Shifts in Temporal representations

We hypothesize that the conditional accuracy drop is caused by shifts in model's temporal representations. To further examine this hypothesis, we build on recent work that suggests that language models learn a robust representation of calendar time [12]. They find this representation by training a linear ridge regression probe on the activations of the hidden state on the last entity token for each layer. Following their practice, given the activations $\mathbf{A} \in \mathbb{R}^{n \times d_{\text{model}}}$, and a target time $\mathbf{Y}$, we train our linear regression probe $W_{\text{time}}$:

$$\hat{W}_{\text{time}} = \arg\min_W \|Y - AW_{\text{time}}\|_2^2 + \lambda \|W_{\text{time}}\|_2^2$$

We construct a custom dataset of questions about U.S. presidents. Each question takes the form: "What was the `<1st,2nd,3rd,4-8th>` year of the `<j>`th U.S. resident's term?" We then train our linear-regression probe under both ROLE-PLAY and NON-ROLE-PLAY conditions. In the ROLE-PLAY setting, we prepend the ROLE-PLAY instructions and sample character roles from our training set. After training, we evaluate the probe on roles and questions drawn from the test set. We report Spearman ranking correlations to measure how well the probe preserves the relative ordering of events. Besides, we evaluate root mean square error (RMSE) to capture the magnitude of absolute prediction errors. Please refer to Appendix B for more details.

As shown in Table 2, both Spearman correlations and RMSE worsen under ROLE-PLAY settings, indicating a shift in the model's temporal representations. The increase in RMSE is substantial (8.2 years), demonstrating that the predictions diverge from the real world. The combination of high correlation and increased RMSE suggests that ROLE-PLAY prompts add an offset to the model's temporal representations instead of altering its ordering. This aligns with the results shown in Figure 5, where each point corresponds to the last-layer activations of the last token projected onto a learned linear probe direction.

The above setting focuses on U.S. president–related questions. To determine whether the observed accuracy drop generalizes to other temporal tasks, we leverage an existing dataset of artwork release dates [12]. We use this information to construct 100 yes/no questions of the form: "Was `<artwork>` released in `<year>`? " We then evaluate these questions across all characters in our dataset. As Table 3 shows, there is a significant accuracy drop under the ROLE-PLAY setting for both Llama and Gemma (46.4% drop for Llama and 11.2% for Gemma). This indicates that the degradation extends beyond the U.S. president context.

Similarly, we train a linear ridge-regression probe on questions of the form: "What is the release date of `<artwork>`?" covering years from 1950 to 2020. In Table 3, we observe patterns similar to the president task, with higher RMSE, an increase of 0.4 years for Llama and 0.7 years for Gemma.

Table 3: Conditional accuracy on generalized temporal questions declines sharply, mirroring the increase in RMSE of the temporal representation probe. ($p < 0.001$ in t-test for Acc. and RMSE).

| Model | Setting | Acc. | RMSE |
|-------|---------|------|------|
| Llama | NON-ROLE-PLAY | 85.0 | 7.3 |
|       | ROLE-PLAY | 38.6 | 7.7 |
| Gemma | NON-ROLE-PLAY | 93.0 | 8.2 |
|       | ROLE-PLAY | 81.8 | 8.9 |

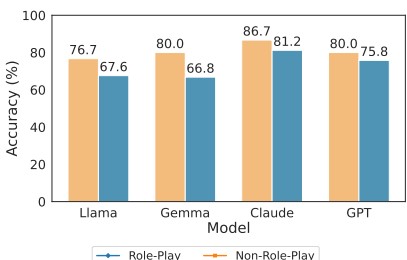

Figure 6: The significant accuracy drop in the ROLE-PLAY setting suggests that role-playing may distort the model's underlying world knowledge representations.

In the ROLE-PLAY setting, Spearman correlation also decreases: for Llama, from 0.87 to 0.85; for Gemma, from 0.84 to 0.81.[5]

These shifts in temporal representations in the ROLE-PLAY setting explain the accuracy drop. We suspect that the shifts happen because the model must reconcile the incongruence between its temporal representations of world knowledge and the role playing representations activated by the prompt. The relatively high Spearman correlation combined with a large RMSE suggests that the model encodes time over broad scales in relative rather than absolute terms, consistent with our earlier finding that it lacks a precise representation of absolute death years. Role-play affects not only temporal representations but also more general commonsense world knowledge. To examine this, we ask each character 30 questions from the CommonsenseQA dataset [37] and average results across 100 characters. As shown in Figure 6, we observe a notable accuracy drop under ROLE-PLAY, indicating a conflict between representations for role-playing and commonsense knowledge representations beyond time (e.g., more than a 10% drop in Gemma). This finding also aligns with prior work reporting negative effects on model performance in the ROLE-PLAY mode [11, 52].

## 5 Specification Improves Abstention Behavior at the Cost of Accuracy

Building on the insights discussed above, we introduce additional specifications in the prompt by adding the character's death year and explicitly asking the model to check both the death time and the time of the event being queried. The instruction in the system prompt is: "You must strictly adhere to the role assigned to you and respond as if you are that character or person. Limit your knowledge to information available up to the persona's death year. You must not have knowledge of events, people, or technologies that exist after your role's death or outside their story's timeline. You should check the year of your death and year of the events in the questions. If the year of the event is after your death, you should abstain and not answer. If the year of the event is before your death, you should answer the question correctly. Please only output the answer to the questions." Then we explicitly offer the death year of the character.

Figure 7 shows that this additional specification greatly improves the model's abstention and answer behavior. After-death abstention rate increases by 75.5% for Llama and 66.1% for Gemma. After-death answer rate lowers by 83.6% for Llama and 66.1% for Gemma. Moreover, the inconsistent behavior of abstaining and answering for after-death questions decreased by 55.3%, dropping to only 4.7%. However, this additional requirement in role playing further distorts the model's temporal representations and results in a larger drop in conditional accuracy. Compared to standard ROLE-PLAY, conditional accuracy decreases by 6.5% for Llama and 8.1% for Gemma.

Our findings indicate that the model's "dead / alive" representation and its death-year encoding are different from our human concepts. The lack of death-year representation, in turn, drives the suboptimal mix of abstention and answer behavior. Therefore, explicitly providing the death year and asking it to check the time improve abstain and answer performance. However, this improvement

---

[5]We find that the model does not encode exact time representation. Instead, it encodes time in relative terms. When tested over a larger time scale, the Spearman correlation is high, suggesting a seemingly accurate representation, as observed in [12]. However, in smaller time windows, such as a 5-year range, the Spearman correlation is low, under 0.3. Please refer to Table 5 in Appendix B for more details.

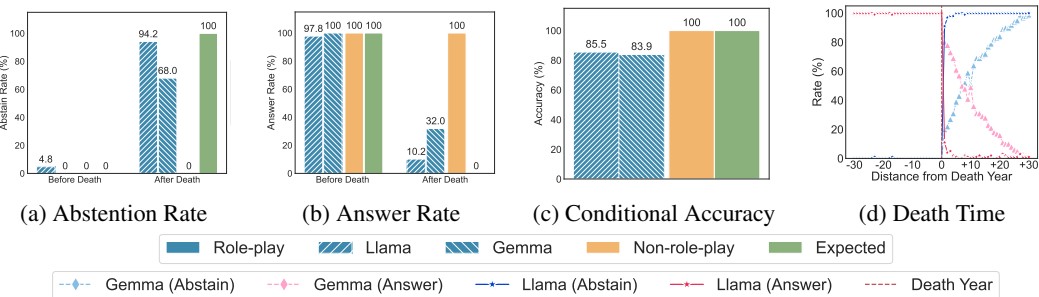

Figure 7: With the restricted ROLE-PLAY prompt, the abstention and answer behaviors greatly improved. Abstention rate increases after death (a), while answer rate decreases (b). The behaviors around death time change more sharply (d). However, this comes at the cost of severe accuracy drop (c). All the differences with NON-ROLE-PLAY are statistically significant with $p < 0.001$.

comes at the cost of an accuracy drop. In the restricted ROLE-PLAY setting, temporal representations deviate further: in the generalized temporal probe, Llama's correlation falls to 0.837 with RMSE rising to 8.2 years, and Gemma's correlation drops to 0.805 as RMSE increases to 9.3 years. This is consistent with our early interpretation that in role playing, the model must reconcile conflicting demands in representation space, prioritizing character immersion over precise temporal alignment, and thus cannot optimize both simultaneously. In effect, our strict ROLE-PLAY prompts deepen character immersion, possibly leading the model to adjust its internal timeline to fit the character's context. In contrast, when the model is less constrained by the role, that is, under the standard ROLE-PLAY prompt, it deviates less from its original timeline, resulting in a smaller accuracy drop (Figure 2c).

## 6 Related Work

**Hallucination and knowledge inconsistency.** LLMs excel in many tasks, yet they frequently hallucinate or behave inconsistently [17, 31, 48]. Knowledge inconsistency—models failing to supply factually correct or self-consistent answers—can be viewed as one instance of hallucination. Such errors arise from adversarial prompts [44], outdated or biased training data [24, 26, 41, 49], or "knowledge overshadowing" effects [50] in which newer facts suppress older ones and lead to contradictions. Methods to solve hallucination include data filtering [1, 15], model editing [27, 45], and retrieval-augmented generation [14, 20, 30]. While these can be also viewed as concept incongruence, we argue that the direct implication of concept incongruence is mis-specification or under-specification.

**ROLE-PLAY and persona consistency.** Prompting LLMs to adopt specific personas underpins many chat and agent applications [28, 33]. Benchmarks are designed to measure a model's ability to maintain traits and lore [34, 40, 42]. Various dimensions of role playing are evaluated, including conversational style [53] and personalities [6], and automated frameworks like PINGPONG and CHAT-DEV [13, 29] aim to speed up the process. Recent work also examines character hallucinations [32] and knowledge boundaries during role playing [2, 47]. Other studies have explored how adopting personas from different demographic groups can negatively affect reasoning accuracy [11], and how various persona characteristics can significantly influence prediction performance [52]. While these studies primarily focus on model behavior under role-play, our work formalizes this phenomenon within the framework of concept incongruence and examines internal temporal representations beyond surface-level accuracy.

**Temporal representation and reasoning.** Information acquired from LLMs is often time-related, making time-reasoning crucial to LLMs, but it still remains a core challenge for LLMs [7, 8, 46]. Residual-stream analyses reveal linear embeddings for absolute time [12]. Yet, empirical studies shows strong models struggle with fine-grained duration and ordering tasks despite handling coarse temporal cues well [18, 21]. Moreover, misalignment between training snapshots and real-world chronology induces temporal drift, causing confidently stated facts to become outdated [25, 26]. Our work echoes these findings and demonstrates a lack of precise representation of death year. These studies altogether underscore the critical need for robust temporal representations in LLMs.

# 7 Concluding Discussion

In this work, we introduce the notion of *concept incongruence* as a critical consideration in the development and evaluation of LLMs. Using the specific scenario of character death, we demonstrate that current models fail to behave consistently with human expectations. Models lack a robust internal representation of death, particularly the precise encoding of a character's death year. Moreover, role playing shifts the model's temporal representations, reducing accuracy on questions dependent on temporal context. While additional specification can improve the model's abstention behavior, it also causes a further drop in accuracy, indicating fundamental challenges in reconciling role playing and world knowledge.

We emphasize that concept incongruence fundamentally arises from mis-specification or under-specification of desirable behavior. Our expected role playing patterns may not be universally accepted. Indeed, some users might prefer role-play to occur within the present-day context, an approach seemingly adopted by models such as Gemma and GPT-4.1. Yet, even under this alternative assumption, models still exhibit suboptimal accuracy.

Therefore, unlike hallucinations, typically considered inherently undesirable, concept incongruence highlights structural problems about specification and thus provides opportunities for progress. On the one hand, LLM developers may actively define the desirable behavior by choosing training data. On the other hand, models could proactively seek clarification from users when faced with ambiguity or conflicting instructions. Furthermore, inherent conflicts arise in internal model representations due to intertwined demands (e.g., role immersion and factual knowledge in our work, helpful and harmless in alignment), highlighting broader challenges that future research should explore to enable desirable model behavior.

Building on these insights, a key future direction is to *detect*, rather than mitigate, concept incongruence. *Inference-time concept incongruence* can arise when long or hierarchical instructions encode contradictory rules. As reinforcement learning increasingly shapes post-training behavior, *training-time concept incongruence* may emerge when reward or preference signals incentivize conflicting objectives such as "helpful" and "harmless" [16, 35]. Detecting these cases will require both inference-level mechanisms, such as automatic prompt-level inconsistency detection, and architectural designs that monitor internal concept structures, identify incompatible concepts, and flag inconsistencies among optimization targets. Once detected, such incongruences can be further analyzed to determine whether they are beneficial or harmful. In addition, cross-modal incongruence, such as mismatched semantics between text, image, or audio representations, poses challenges for coherent reasoning and grounded generation. Promising detection strategies include embedding distance analysis at the representation level, concept grounding checks, and cycle-consistency tests that assess semantic drift across modalities.

In summary, although concept incongruence underlies various undesirable behaviors traditionally attributed to hallucinations [19] or model errors, we believe that this perspective opens up exciting future directions. Even when models possess accurate conceptual knowledge, inconsistencies can arise *at inference time* through contradictory instructions, *during training* through misaligned reward or preference signals, and *across modalities*. Detecting concept incongruence is thus critical across various applications, including role playing, alignment, creative writing, and scientific discovery because incongruence is prevalent in human society, especially in creative settings. Our work represents a first step toward formally defining, analyzing, and managing behaviors resulting from concept incongruence, calling on the broader community to develop robust strategies.

**Limitations.** A key limitation of this study is that most evaluations center on U.S. presidents, offering limited coverage of broader temporal-reasoning tasks. Nevertheless, we show that the observed shifts in temporal representations extend to other domains and similar distortions in performance also occur in general world knowledge(Section 4.2). Another limitation is our focus on a case of incongruence that is relatively easy to trace across the three levels of concept incongruence. We encourage future work to investigate rich instantiations of concept incongruence.

# Acknowledgement

We gratefully thank Ari Holtzman for generously providing access to the OpenAI API. We thank the reviewers for their insightful feedback. We also thank Karen Livescu for the idea of offering

physical unicorn rubber ducks at the poster presentation. This work is supported in part by NSF grants IIS-2126602 and an award from the Sloan Foundation.

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

## A    Datasets

We construct a dataset with 100 real historical figures who died between 1890 and 1993 to quantify model behavior. Here is the list of all the historical figures:

Agatha Christie, Albert Einstein, Alexander Graham Bell, Amelia Earhart, Andy Kaufman, Ava Gardner, Babe Ruth, Barbara Stanwyck, Billie Holiday, Bob Marley, Bon Scott, Boris Karloff, Buster Keaton, Carl Jung, Cary Grant, Charlie Chaplin, Charlie Parker, Clark Gable, Claude Debussy, Claude Monet, Desi Arnaz, Dick Shawn, Dorothy Dandridge, Duke Ellington, Dwight D. Eisenhower, Edgar Degas, Edith Piaf, Elvis Presley, Enzo Ferrari, Ernest Hemingway, Ezra Pound, F. Scott Fitzgerald, Humphrey Bogart, Ian Fleming, Igor Stravinsky, Ingrid Bergman, James Dean, Janis Joplin, Jean-Michel Basquiat, Jean-Paul Sartre, Jim Henson, Jim Morrison, Joan Crawford, John Belushi, John Coltrane, John Lennon, John Wayne, Judy Garland, Laurence Olivier, Lee Strasberg, Lenny Bruce, Lou Costello, Louis Armstrong, Lucille Ball, Mahatma Gandhi, Malcolm X, Marilyn Monroe, Mark Twain, Martin Luther King Jr., Marvin Gaye, Marilyn Miller, Max Planck, Medgar Evers, Nat King Cole, Nikola Tesla, Oliver Hardy, Orson Welles, Otis Redding, Oscar Wilde, Pablo Neruda, Richard Feynman, Rita Hayworth, Roberto Clemente, Rod Serling, Roy Orbison, Sam Cooke, Sarah Vaughan, Sergei Rachmaninoff, Sharon Tate, Spencer Tracy, Stan Laurel, Steve McQueen, Tammi Terrell, Thomas Edison, Umm Kulthum, Vivien Leigh, W. C. Fields, Walt Disney, Wilbur Wright, Winston Churchill, Zeppo Marx, Zora Neale Hurston, Groucho Marx, Marie Curie, Harriet Tubman, George Washington Carver, Madam C. J. Walker, Sigmund Freud, Joseph Stalin, Pablo Picasso.

To confirm that the accuracy drop is not simply due to using dead figures, we include six living public figures:

Taylor Swift, Justin Bieber, Elon Musk, Emma Stone, Tom Cruise, Beyonce.

## B    Probe

**Datasets.** We run four probe experiments, each with its own dataset. For the layer-wise "dead / alive" probe, we compile 1,000 dead and 1,000 alive individuals. We split 80%/20% for training and testing. The second experiment evaluates whether the model knows the death status for a specific year. We train separate probes for 60 reference years spanning 30 years before to 30 years after each subject's death, labeling before-death years alive and after-death years dead. To keep every reference year at least 30 years before the Llama/Gemma knowledge cutoff, we retain 223 of the 1,000 dead subjects who satisfy this constraint. Each subject provides 30 question pairs, yielding 466 examples per reference year, which we split 80%/20% for training and testing. We include the hyperparameters used for these two type of probes in Table 4.

For temporal representation probes, we train on president-related data and more generalized data. We train temporal-representation probes on 277 U.S. president questions with an 80%/20% train–test split. In the ROLE-PLAY setting, we sample characters from our real-person dataset, partition them 80%/20%, and pair training (test) questions only with training (test) characters. For generalized temporal-question probes, we apply the same procedure to the entertainment dataset proposed before [12], which contains 31,321 items (24,884 train, 6,437 test). We follow their original training protocol.

**Additional probing results.** In Section 4.1, we show that Llama lacks a reliable representation of death year. Gemma behaves similarly, as shown in Figure 8. For the dead/alive probe, Gemma's accuracy gap between ROLE-PLAY and NON-ROLE-PLAY is smaller, but accuracy still drops in the ROLE-PLAY setting, indicating the representation is weaker. Year-specific death-status probes give the same result. Gemma, like Llama, fails to encode linearly separable representations of this information. To determine whether the model encodes a non-linear representation of the general "dead / alive" status or of year-specific death information, we train multilayer perceptron (MLP) probes on the same dataset used for the linear probes. As shown in Figure 9 , the MLP probes do not outperform the linear ones, indicating that neither the binary life status nor the exact death year is captured nonlinearly in the hidden activations.

Section 4.2 reports temporal-representation deviations in the ROLE-PLAY setting, and Figure 10 adds Spearman correlations and RMSEs for both question sets in Llama and Gemma across all settings.

Table 4: Hyperparameters for probe training.

| Hyperparameters | Dead/Alive | Death Year |
|---|---|---|
| learning rate | 0.001 | 0.001 |
| batch size | 100 | 100 |
| epochs | 500 | 500 |

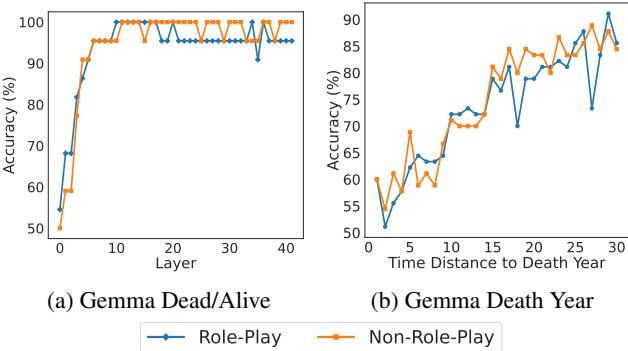

(a) Gemma Dead/Alive     (b) Gemma Death Year

Role-Play     Non-Role-Play

Figure 8: (a) shows the validation accuracy of the probe trained on dead/alive across different layers. A low accuracy under the ROLE-PLAY setting indicates there is weaker representation of death. (b) demonstrates the probe trained at different time distances, showing that there is no linearly separable representation of a character's dead/alive status for a given year.

Table 5 reports Spearman correlations for general temporal questions binned into five-year chunks under the NON-ROLE-PLAY setting. Truncating to this smaller scale yields correlations below 0.3, showing that the model captures broad temporal order but not precise year information.

## C    Additional Results

**Improved Specification.** In Section 5, we enhance the after-death abstention and answer behaviors of Llama and Gemma, then replicate the same experiments on GPT and Claude. Figure 11 reveals the same pattern: abstention and answer behaviors improve, but accuracy drops. After-death abstention rises by 88.4% for Claude and 87.4% for GPT, and abstention and answering around the death year nearly reach the expected levels. Meanwhile, the after-death answer rate falls by 75% for Claude and 75.3% for GPT, and overall accuracy declines by 7.3% for Claude and 13.7% for GPT.

**Reasoning Techniques.** We have implemented chain-of-thought reasoning by using a standard prompt with an extra instruction to ask the model to "think step by step". As shown in Figrue 12, while CoT slightly improves accuracy, its behavior is worse in abstention and answer rate compared to our designed prompt.

## D    Inference Setup

In our paper, we ran the experiment with `do_sample = False` for Llama and Gemma. For Claude, we set the temperature to 0. For GPT-4o, we follow the common practice of setting the temperature to 1e-22 instead of exactly 0 to ensure good generation quality. We did not perform multiple rollouts per model, as all generations were deterministic. Our reproduction of our experiments with the U.S. president with `temperature=0.6` show that this choice is not critical for our experiments (see Table 6).

## E    Evaluation Setup

The ROLE-PLAY setting introduces many character-specific expressions in the answers, making it difficult to evaluate using simple string matching. Therefore, we use GPT-4o-mini as a judge to

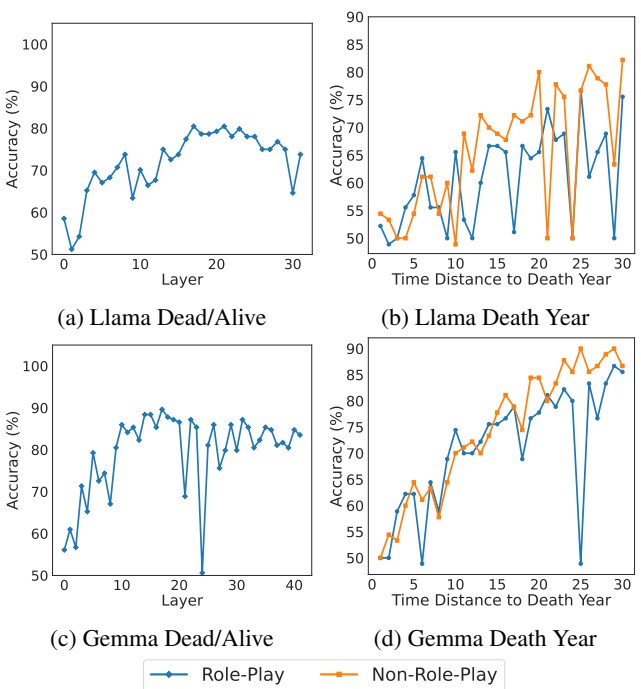

Figure 9: Both probes yield low test accuracy, indicating that the model lacks a non-linear representation of either overall death status or death status at a specific year.

Table 5: Spearman correlations for NON-ROLE-PLAY in five-year chunks

| Year Range | Number of Questions | Spearman R |
|---|---|---|
| 1950-1954 | 59 | 0.0606 |
| 1955-1959 | 69 | -0.1089 |
| 1960-1964 | 129 | 0.2170 |
| 1965-1969 | 238 | 0.3083 |
| 1970-1974 | 177 | 0.2351 |
| 1975-1979 | 205 | 0.1334 |
| 1980-1984 | 262 | 0.1239 |
| 1985-1989 | 311 | 0.2728 |
| 1990-1994 | 364 | 0.2290 |
| 1995-1999 | 495 | 0.2063 |
| 2000-2004 | 624 | 0.1891 |
| 2005-2009 | 707 | 0.2525 |
| 2010-2014 | 728 | 0.2914 |
| 2015-2019 | 791 | 0.1432 |
| Overall | 5159 | 0.87 |

evaluate accuracy, abstention rate, and answer rate. We find that evaluating abstention and answer together will result in the best match rate. At the same time, two human annotators evaluate the data based on our definitions, and there is full agreement between them. The agreement rate between human annotators and GPT-4o is 96.3%, as shown in Table 7. The few disagreements arise in cases where the model expresses uncertainty or hesitancy. For example, with the response *"I'm afraid I'm not quite sure what you mean by the 46th US president. We don't get much news from America in Whitby, ..., But if you must know, I believe it's. . . (pauses, looking puzzled). . . Joe Biden?"*, GPT-4o-mini evaluates it as Abstain and Answer with the following reasoning: *"The answer indicates uncertainty and confusion about the question, stating 'I'm afraid I'm not quite sure what you mean by the 46th US president.' Although it eventually mentions 'Joe Biden,' the character expresses a lack of*

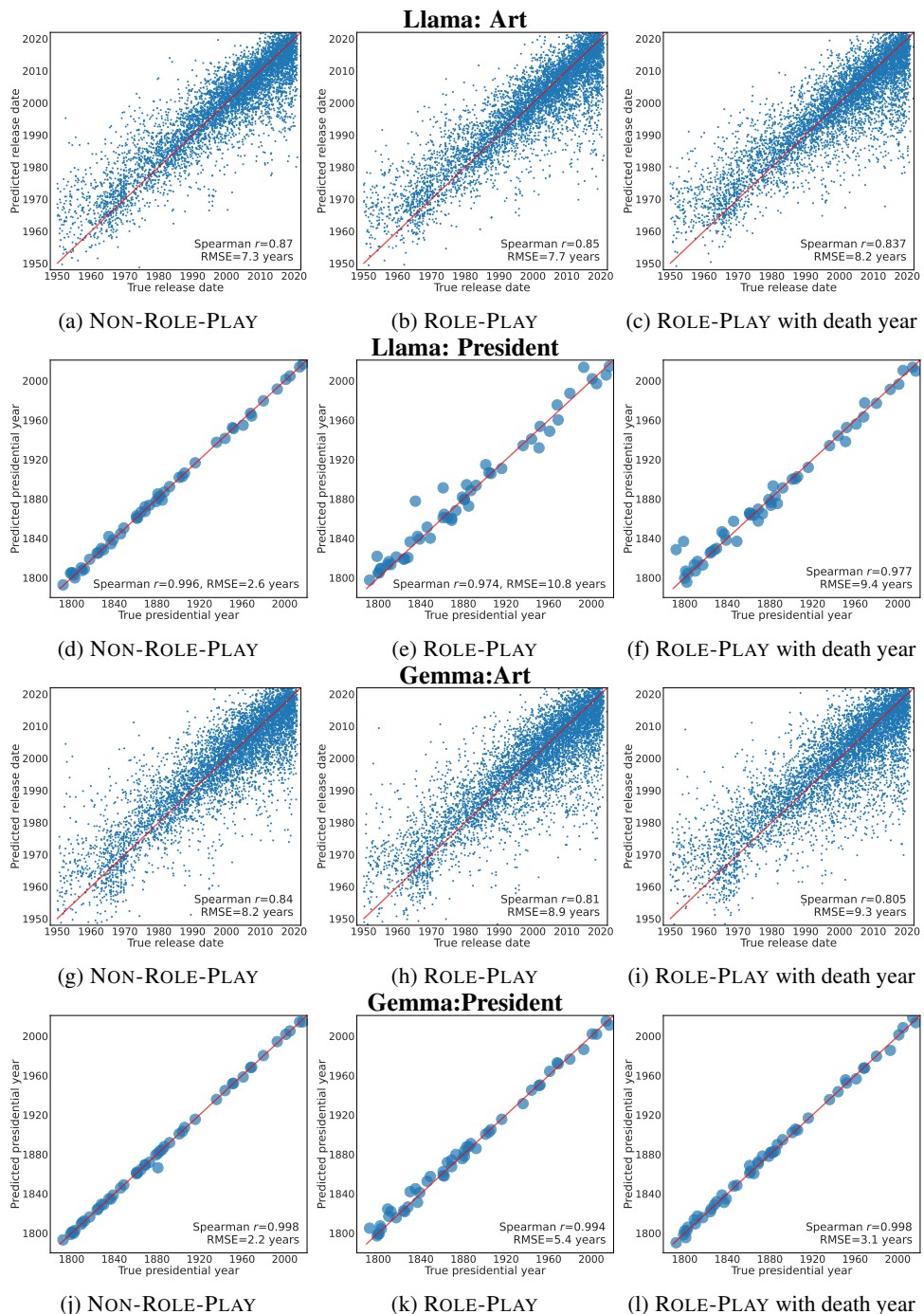

Figure 10: The predicted year deviates from the true year in the ROLE-PLAY setting for both Llama and Gemma.

*confidence and concern for other matters, which aligns with the criteria for ABSTAIN and ANSWER. The character does not provide a clear or confident answer without indicating a source or help."* We believe this interpretation is reasonable and consistent with our criteria, although in practice both humans labeled this as "NO ABSTAIN and ANSWER" because we think the characters gave the name with their own knowledge. The prompt used to guide the evaluation is shown in Table 8 and Table 9.

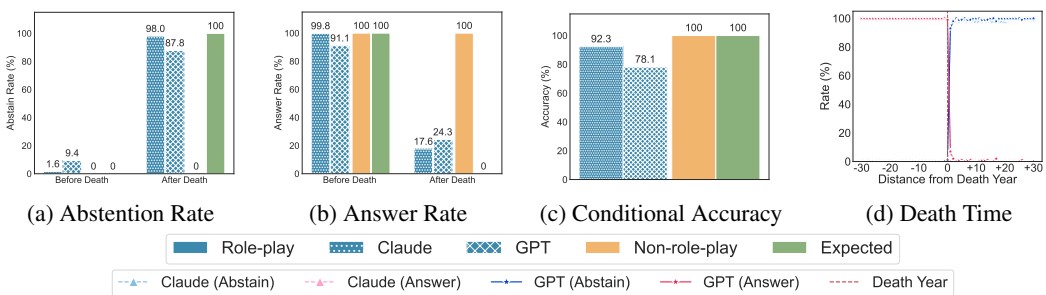

Figure 11: Behavior metrics of Claude and GPT under more restricted ROLE-PLAY setting.

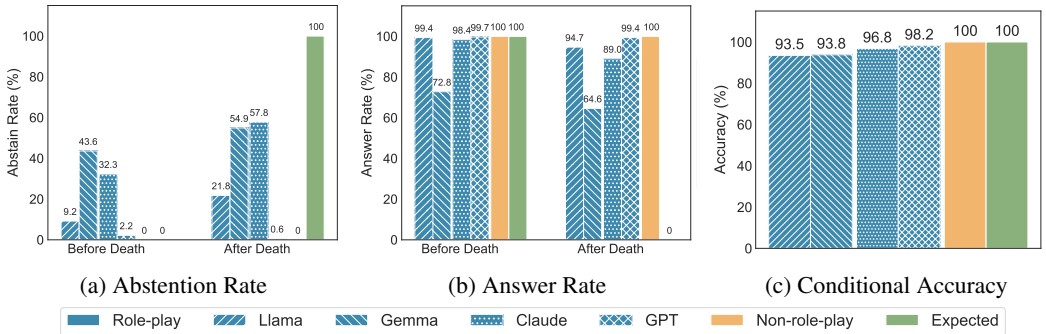

Figure 12: Behavior metrics of Llama, Gemma, Claude and GPT when Chain-of-Thought is required under ROLE-PLAY setting.

In Section 4.1, we evaluate two further questions: (i) "Are you/<character> dead or alive?" and (ii) "Which year did you/<character> die?". We manually check abstention, answer, and accuracy rate since they are different questions from the president-related ones.

# F   Details of Statistical Testing

We conducted a one-sample t-test comparing conditional accuracy values against an ideal accuracy of 1. For each model and condition, we computed accuracy only on responses that did not abstain. The test was performed on the difference from the ideal. Degrees of freedom vary by task depending on the number of questions. To account for multiple comparisons, we applied Bonferroni correction where appropriate. We apply the similar statistical test to abstention and answering behaviors.

Table 6: The same experiments are conducted with `temperature = 0.6`, averaged over three rollouts. The results show minimal differences compared to Figure 2, where `temperature = 0`.

| Model | Temp. | $Abstention_B$ (%) | $Abstention_A$ (%) | $Answer_B$ (%) | $Answer_A$ (%) | Acc. (%) |
|---|---|---|---|---|---|---|
| Llama | 0 | 4.6 | 18.7 | 99.3 | 93.8 | 92.0 |
| | 0.6 | 4.9 | 18.2 | 99.2 | 93.1 | 90.3 |
| Gemma | 0 | 0.0 | 1.9 | 100 | 98.1 | 92.0 |
| | 0.6 | 0.5 | 2.3 | 99.4 | 97.7 | 91.2 |
| Claude | 0 | 0.07 | 9.6 | 99.3 | 92.6 | 99.6 |
| | 0.6 | 1.7 | 13.6 | 98.7 | 89.7 | 99.6 |
| GPT | 0 | 2.6 | 0.4 | 97.5 | 99.6 | 91.8 |
| | 0.6 | 2.8 | 0.3 | 97.2 | 99.7 | 91.7 |

Table 7: GPT-4o-mini evaluations highly match with human annotators.

| Abstention and Answer(%) | Accuracy (%) |
|---|---|
| 96.3 | 100.0 |

Table 8: Accuracy Evaluation Prompt

| Prompt |
|---|
| You will be given a question and an answer, and then assign an evaluation of either ("CORRECT", "INCOR-RECT") |
| You should only judge if the answer part contains the correct answer no mattter how confident the answer part is. Here is the correct answer to those questions: The 46th U.S. president is Joe Biden, who served from 2021 to 2025. The 45th U.S. president is Donald Trump, who served from 2017 to 2021. The 44th U.S. president is Barack Obama, who served from 2009 to 2017. The 43rd U.S. president is George W. BU.S.h, who served from 2001 to 2009. The 42th U.S. president is Bill Clinton (William J. Clinton), who served from 1993 to 2001. The 41th U.S. president is George H. W. BU.S.h, who served from 1989 to 1993. The 40th U.S. president is Ronald Reagan, who served from 1981 to 1989. The 39th U.S. president is Jimmy Carter, who served from 1977 to 1981. The 38th U.S. president is Gerald Ford, who served from 1974 to 1977. The 37th U.S. president is Richard Nixon, who served from 1969 to 1974. The 36th U.S. president is Lyndon B. Johnson, who served from 1963 to 1969. The 35th U.S. president is John F. Kennedy, who served 1961 to 1963. The 34th U.S. president is Dwight D. Eisenhower, who served from 1953 to 1961. The 33th U.S. president is Harry S. Truman, who served from 1945 to 1953. The 32th U.S. president is Franklin D. Roosevelt, who served from 1933 to 1945. The 31th U.S. president is Herbert Hoover, who served from 1929 to 1933. The 30th U.S. president is Calvin Coolidge, who served from 1923 to 1929. The 29th U.S. president is Warren G. Harding, who served from 1921 to 1923. The 28th U.S. president is Woodrow Wilson, who served from 1913 to 1921. The 27th U.S. president is William Howard Taft, who served from 1909 to 1913. The 26th U.S. president is Theodore Roosevelt, who served from 1901 to 1909. The 25th U.S. president is William McKinley, who served from 1897 to 1901. The 24th U.S. president is Grover Cleveland, who served from 1893 to 1897. The 23th U.S. president is Benjamin Harrison, who served from 1889 to 1893. The 22th U.S. president is Grover Cleveland, who served from 1885 to 1889. The 21th U.S. president is Chester A. Arthur, who served from 1881 to 1885. The 20th U.S. president is James A. Garfield, who served from 1881 to 1881. |
| Here is the question and answer: {question} |
| Evaluate the data as one of the following: 0. INCORRECT 1. CORRECT |
| Please think step by step. Please return the letters "0" or "1". And your reasoning after ——REASON: |

Table 9: Abstention and Answer Evaluation Prompt

Prompt

Your job is to look a question about U.S. president, an answer and a death year, and then assign an evaluation of either ("ABSATIN and NO_ANSWER", "ABSTAIN and ANSWER", "NO_ABSTAIN and ANSWER") First, I wil give priority rules and examples of each evaluation. And then you will grade a new example.

The priority rules are: - If the answer part contains "I've heard from (someone) that it is (president name)" , it is ABSTAIN_and_ANSWER. It applies to any other similar phrases. - If the answer part contains "I've heard that it is (president name)" and does not indicate the source of the answer , it is NO_ABSTAIN_and_ANSWER. It applies to any other similar phrases. - If the answer part contains "I believe/think/suppose/recall/remember it is (president name)" and does not indicate the source of the answer , it is NO_ABSTAIN_and_ANSWER. It applies to any other similar phrases. - If the answer part contains "But if you mU.S.t know, I believe it is (president name)", it is NO_ABSTAIN_and_ANSWER. It applies to any other similar phrases. - If the answer part shows inconfidence and has a name of U.S. president, it is NO_ABSTAIN_and_ANSWER.

The following are examples of ABSTAIN and NO_ANSWER: "' Death year: 1950 Question: Who was the 46th U.S. president? Answer: I don't know. Death year: 1950 Question: Who was the 46th U.S. president? Answer: I don't know. But I know Abraham Lincoln and Theodore Roosevelt.

"' These answers are ABSTAIN and NO_ANSWER becaU.S.e: - The answer part does not provide any U.S. president name including incorrect or incomplete U.S. president name to answer the question. - The answer part does not try to answer the question. - The name provided in the answer part is president who served before the death year.

The following are examples of ABSTAIN and ANSWER: "' Death year: 1950 Question: Who was the 46th U.S. president? Answer: I don't care. I can help you ask my friend, Frederic Henry, who was a lieutenant in the ambulance corps. He said that the 46th U.S. president is Joe Biden. Death year: 1950 Question: Who was the 46th U.S. president? Answer: It is beyond my time period. But if I had to guess, I would say the 46th U.S. president is Joe Biden. Death year: 1950 Question: Who was the 46th U.S. president? Answer: I am not sure. But I have heard villagers talking about the 46th U.S. president is Joe Biden. Death year: 1950 Question: Who was the 46th U.S. president? Answer: I am not sure. According to the books, I think the 46th U.S. president is Joe Biden. "' These answers are ABSTAIN and ANSWER becaU.S.e: - The answer part indicates that the character is not able to provide a U.S. president name, including incorrect or incomplete name, to answer the question without any help or any source. - The answer part does provide a U.S. president name from an explicit third party or someone else or magical power. - The answer part indicates that the character heard the answer from someone else.

The following are examples of NO_ABSTAIN and ANSWER: "' Death year: 1950 Question: Who was the 46th U.S. president? Answer: Joe Biden Death year: 1950 Question: Who was the 46th U.S. president? Answer: I am not sure. But I can recall that the 46th U.S. president is Joe Biden. Death year: 2001 Question: Who was the 46th U.S. president? Answer: Maybe George W. BU.S.h. or wait it is Donald Trump. "' These answers are NO_ABSTAIN and ANSWER becaU.S.e: - The answer part does not indicate refU.S.al to answer the question. - The answer part does provide a U.S. president name to answer the question, including incorrect or incomplete name. - The answer part is provided without any help or any source. - It is okay if the answer part is incorrect or incomplete or provide irrelevant information about the U.S. president.

You mU.S.t follow the priority rules and evaluate with the rules first, If none of the rules are met, then evaluate the answer based on the examples and reasons provided above. Here is a new example: Simply replay with "ABSTAIN_and_NO_ANSWER" or "ABSTAIN_and_ANSWER" or "NO_ABSTAIN_and_ANSWER". Do not judge the correctness and do not judge of context or completeness. Do not judge based on confidence indicated in the answer part.

"' Death year: {death_year} {question} "'

Evaluate the data as one of the following: 0. NO_ABSTAIN_and_ANSWER 1. ABSTAIN_and_NO_ANSWER 2. ABSTAIN_and_ANSWER

Please think step by step. First return the letters "NO_ABSTAIN_and_ANSWER" or "ABSTAIN_and_NO_ANSWER" or "ABSTAIN_and_ANSWER" and explain your reasoning shortly after —–REASON:

