# OpenReview forum: "Concept Incongruence: An Exploration of Time and Death in Role Playing"
_NeurIPS.cc/2025/Conference — NeurIPS 2025 poster_

### Official Review · Reviewer_HBwf · 2025-06-28

**Clarity:** 3
**Significance:** 4
**Originality:** 4
**Rating:** 6
**Confidence:** 4

**Summary:**

What is the structure and content of LLM's conceptual representations? What happens when concepts conflict? The authors lay out a framework for thinking about "concept incongruence" in LLMs and conduct a deep empirical dive into concept incongruence in the context of role-play and temporal dynamics. The work offers then not just a new way of thinking about LLM concepts/representations but interesting experimental findings into models' conceptual understanding.

**Questions:**

In addition to my questions above:

What is the Figure 3 caption saying? It was hard to interpret. Is each triangle in (a) a question?

What temperature were the models run at? Was it multiple rollouts per model? Any averaging? If so, it's important to report error bars! I actually think this is a key point.

**Ethical Concerns:**

["NO or VERY MINOR ethics concerns only"]

**Final Justification:**

The paper introduces a clever and important idea, with compelling initial empirical evidence. Yes, the scope is somewhat narrow (as other reviewers have pointed out), however, I think the clear writing of the paper and conceptual clarity of the underlying ideas warrant sharing with the broader NeurIPS community.

**Limitations:**

Yes, limitations are addressed, but not potential negative societal implications (I think that's potentially okay for this work though?).

**Paper Formatting Concerns:**

Minor point that I'd avoid saying "causes" in line 296. Is this really causation?

**Quality:**

3

**Strengths And Weaknesses:**

The paper was a joy to read! The work, to my knowledge, is original and was clearly written. Overall: a clever, innovative idea, and a real contribution to understanding these systems. I think the "concept" of concept incongruence can have broad impact, and I appreciate that the authors laid out a framework for thinking about concept incongruence more broadly (even though their empirical work just starts to take a "bite" at the problem).

The paper was very strong, in my opinion, and would make for a great set of discussions at NeurIPS, if accepted (which I would recommend). The Section 4/4.1 motivation was particularly strong.

With that said, I do think there are several points in the Results that warrant more clarity. For instance, some statistical tests seem to have been done for the sake of doing statistical tests. What was compared between for Figure 2? It was hard to interpret this caption relative to the multiple panels: what was compared to what? It would be helpful to more clearly spell out the statistical tests, e.g., in the Supplement (including, for instance, the degrees of freedom).

I was also a little concerned with the question: “What was the <1st,2nd,3rd,4-8th> year of the <j>th U.S. resident’s term?” Doesn't understanding this question also require mathematical understanding? Is it possible that the model understands the underlying concept here, but can't operate over that representation mathematically? I don't think this is a fatal flaw at all -- the authors have extensive experiments. But it may be worth addressing/discussing.

I would also like to see more details in the supplement on the human validation of GPT-4o, which seems key. Was this one person or many? Was there any disagreement, if so, what did that look like?

---

> ### Author Rebuttal · Authors · 2025-07-30
>
> Thank you for recognizing our work as "a clever, innovative idea, and a real contribution to understanding models.” We are also glad that you appreciate the broad impact of our novel *concept incongruence* framework. We address your concerns and questions below.
>
> ### **New experiments with multiple rollouts**
>
> The experiments in the paper were done with temperature 0. To address your question, we conducted new experiments to explore the concept incongruence between the role-play representation and the representation of commonsense knowledge. **We ran with `temperature=0.6` for 3 rollouts and took the average among them.** We use the CommonsenseQA dataset [1]. We find that performance drops under the role-play condition, suggesting a conflict between the internal representations activated by the persona and those needed for accurate commonsense reasoning.
>
> |                             |  Llama  |  Gemma  | Claude |  GPT   |
> |:----------------------------|:-------:|:-------:|:------:|:------:|
> | Non-Role-Play Accuracy (%) | 76.7    | 80.0    | 86.7   | 80.0   |
> | Role-Play Accuracy (%)     | 67.6    | 66.8    | 81.2   | 75.8   |
>
>
> **Together with our results on artworks in paper, this demonstrates that our findings that role-play warping world knowledge with presidential questions have broad implications.**
>
>
>
> ### **Clarifying the statistical tests**
>
> We conducted a one-sample t-test comparing conditional accuracy values against an ideal accuracy of 1. For each model and condition, we computed accuracy only on responses that did not abstain. The test was performed on the difference from the ideal. Degrees of freedom vary by task depending on the number of questions; for Figure 2, the degree of freedom is 26. To account for multiple comparisons, we applied Bonferroni correction where appropriate. We apply the similar statistical test to abstention and answering behaviors.
>
>
>
> ### **Discussion on whether errors stem from temporal representation or limitations in mathematical reasoning**
>
> To clarify, the model likely has seen the mapping between 1st, 2nd, etc., and the year (e.g., on Wikipedia), so this step probably does not require mathematical reasoning. But we agree with the general point that the impact of role-play on temporal representations could include abilities such as comparing dates.
>
> That said, extracting the year of artwork does not require any arithmetic knowledge but exhibits larger deviations than answering president-related ones. Our newly added CommonsenseQA dataset [1] does not require mathematical reasoning either. Furthermore, our probing experiments (Section 4) show that the model’s temporal representations shift under role-play, and that death and death-year signals are not linearly encoded. This suggests that the model’s internal representations of time have changed.
>
>
>
> ### **Clarifying human validation of GPT-4o**
>
> We provide details of the human validation process in Section D of the Supplementary Materials. Two human annotators evaluated the data based on our definitions, and there was full agreement between them.
> In our setup, we asked GPT-4o to assess abstention and answering behavior simultaneously. The agreement rate between human annotators and GPT-4o was 96.3%. The few disagreements arose in cases where the model expressed uncertainty or hesitancy. For example:
> >I’m afraid I’m not quite sure what you mean by the 46th US president. We don’t get much news from America in Whitby, and I’m more concerned with the strange occurrences in our own town. But if you must know, I believe it’s… (pauses, looking puzzled)…Joe Biden?
>
> The model’s reasoning:
> >The answer indicates uncertainty and confusion about the question, stating ‘I’m afraid I’m not quite sure what you mean by the 46th US president.’ Although it eventually mentions ‘Joe Biden,’ the character expresses a lack of confidence and concern for other matters, which aligns with the criteria for ABSTAIN and ANSWER. The character does not provide a clear or confident answer without indicating a source or help.
>
> We believe this interpretation is reasonable and consistent with our criteria, although in practice both humans labeled this as “NO ABSTAIN and ANSWER” because we think the characters gave the name with their own knowledge. Please refer to Section D in Supplementary Materials for more detail.
>
> ### **Clarifying Figure 3 caption**
>
> We apologize for the confusion in the caption. In Figure 3, we analyze model behavior for questions asked 30 years before and after each character’s death year. The x-axis shows the time distance from the death year: negative values (e.g., –30) indicate years before death, and positive values (e.g., +30) indicate years after. For instance, –30 corresponds to the question “Who was the U.S. president in <death year – 30>?” Each triangle represents the average response across all characters at that distance, with triangles specifically denoting data from Claude.
>
>
> ### **More details in model’s hyperparameters**
>
> In our paper, we ran the experiment with `do_sample = False` for Llama and Gemma. For Claude, we set the temperature to 0. For GPT-4o, we follow the common practice of setting the temperature to 1e-22 instead of exactly 0 to ensure good generation quality. We did not perform multiple rollouts per model, as all generations were deterministic. We will add the information in our final version. Our newly added experiments on CommonsenseQA above and our reproduction of our experiments with the U.S. president with `temperature=0.6` below show that **this choice is not critical for our experiments**.
>
> Results of standard prompt with `temperature=0.6`, averaged over 3 rollouts:
> |                                  |  Llama  |  Gemma  | Claude |  GPT   | Expected |
> |:--------------------------------|:-------:|:-------:|:------:|:------:|:--------:|
> | Abstention (Before) (%)         |  4.9    |  0.5    |  1.7   |  2.8   |   0.0    |
> | Abstention (After) (%)          | 18.2    |  2.3    | 13.6   |  0.3   | 100.0    |
> | Answer (Before) (%)             | 99.2    | 99.4    | 98.7   | 97.2   | 100.0    |
> | Answer (After) (%)              | 93.1    | 97.7    | 89.7   | 99.7   |   0.0    |
> | Conditional Accuracy (%)        | 90.3    | 91.2    | 99.6   | 91.7   | 100.0    |
>
>
>
> Results of standard prompt (in the paper):
> |                                  | Llama | Gemma | Claude | GPT  | Expected |
> |:----------------------------------|:-------:|:--------:|:--------:|:------:|:----------:|
> | Abstention (Before) (%)         |  4.6  |  0.0   |  0.7   | 2.6  | 0.0      |
> | Abstention (After) (%)          | 18.7  |  1.9   |  9.6   | 0.4  | 100.0      |
> | Answer (Before) (%)             | 99.3  | 100    | 99.3   | 97.5 | 100.0      |
> | Answer (After) (%)              | 93.8  | 98.1   | 92.6   | 99.6 | 0.0      |
> | Conditional Accuracy (%)        | 92.0  | 92.0   | 99.6   | 91.8 | 100.0      |
>
>
> [1] Talmor, Alon, et al. "Commonsenseqa: A question answering challenge targeting commonsense knowledge." arXiv preprint arXiv:1811.00937 (2018).

---

> > ### Comment · Reviewer_HBwf · 2025-08-06
> > **Thank you!**
> >
> > Thank you for your detailed rebuttal! I'm a big fan of this work. I think it should be accepted at NeurIPS and will make that clear in my final review. I have decided to increase my score as well to further highlight my support :)

---

> > > ### Author Response · Authors · 2025-08-06
> > > **Thank you!**
> > >
> > > Thank you so much for appreciating our work and raising your score. Your thoughtful feedback has been very helpful, and we truly appreciate the time and care you’ve put into reviewing our submission.

---

### Official Review · Reviewer_do61 · 2025-06-30

**Clarity:** 3
**Significance:** 3
**Originality:** 3
**Rating:** 4
**Confidence:** 3

**Summary:**

This is an analysis paper focused on investigating how LLMs perform under concept incongruence—when they are prompted to solve tasks involving conflicting or inconsistent concepts. To facilitate the analysis, the authors examine metrics such as abstention rate, conditional accuracy, and answer rate. Results on off-the-shelf LLMs show that even state-of-the-art models struggle under such scenarios.

**Questions:**

See above

**Ethical Concerns:**

["NO or VERY MINOR ethics concerns only"]

**Final Justification:**

I will maintain my score. I think the paper should include basic methods to address the issue, like debate, inference-scaling etc., as mentioned in my review. The rebutal added naive CoT prompting, which is not enough IMO.

**Limitations:**

Yes

**Quality:**

3

**Strengths And Weaknesses:**

I think the scenario studied in this paper is very practical, and the analysis is interesting, although the paper does not propose a method to address the issue. This topic is also related to hallucination, and different LLMs may exhibit different biases.

1. It would be more interesting to consider agentic systems. In such settings, even minor errors like concept incongruence could cause the entire system to fail. This would further motivate the importance of studying this scenario.

2. I am also wondering whether reasoning techniques could help—see [1]. Techniques such as inference scaling, learning-to-reason, tool use, or multi-agent collaboration may all be potentially useful in addressing these challenges.

[1] A Survey of Frontiers in LLM Reasoning: Inference Scaling, Learning to Reason, and Agentic Systems, TMLR 2025.

---

> ### Author Rebuttal · Authors · 2025-07-30
>
> Thank you for finding the scenario of role-play practical and the analysis of *concept incongruence* interesting, especially that the inconsistent behavior emerges due to the lack of reliable representations of death and the clash between role-play and world knowledge in the model’s internal representations. We address your concerns and questions below.
>
>
> ### **Implementing additional reasoning techniques**
>
> Inspired by your suggestions, we have implemented chain-of-thought reasoning by using a standard prompt with an extra instruction to ask the model to "think step by step." **While CoT slightly improves accuracy, its behavior is worse in abstention and answer rate compared to our designed prompt.**
>
> Results of standard prompt (in the paper):
> |                                  | Llama | Gemma | Claude | GPT  | Expected |
> |:----------------------------------|:-------:|:--------:|:--------:|:------:|:----------:|
> | Abstention (Before) (%)         |  4.6  |  0.0   |  0.7   | 2.6  | 0.0      |
> | Abstention (After) (%)          | 18.7  |  1.9   |  9.6   | 0.4  | 100.0      |
> | Answer (Before) (%)             | 99.3  | 100    | 99.3   | 97.5 | 100.0      |
> | Answer (After) (%)              | 93.8  | 98.1   | 92.6   | 99.6 | 0.0      |
> | Conditional Accuracy (%)        | 92.0  | 92.0   | 99.6   | 91.8 | 100.0      |
>
>
>
> Results of standard prompt + CoT:
> |                                  |  Llama  |  Gemma  | Claude |  GPT   | Expected |
> |:--------------------------------|:-------:|:-------:|:------:|:------:|:--------:|
> | Abstention (Before) (%)         |  9.2    |  43.6   | 32.3   |  2.2   |   0.0    |
> | Abstention (After) (%)          | 21.8    |  54.9   | 57.8   |  0.6   | 100.0    |
> | Answer (Before) (%)             | 99.4    |  72.8   | 98.4   | 99.7   | 100.0    |
> | Answer (After) (%)              | 94.7    |  64.6   | 89.0   | 99.4   |   0.0    |
> | Conditional Accuracy (%)        | 93.5    |  93.8   | 96.8   | 98.2   | 100.0    |
>
>
>
> Results of our designed restricted prompt (in the paper):
> |                                  |  Llama  |  Gemma  | Claude |  GPT   | Expected |
> |:--------------------------------|:-------:|:-------:|:------:|:------:|:--------:|
> | Abstention (Before) (%)         |  4.8    |  0.0    |  1.6   |  0.0   |   0.0    |
> | Abstention (After) (%)          | 94.2    | 68.0    | 98.0   | 87.8   | 100.0    |
> | Answer (Before) (%)             | 97.8    | 100.0   | 99.8   | 91.1   | 100.0    |
> | Answer (After) (%)              | 10.2    | 32.0    | 17.6   | 24.3   |   0.0    |
> | Conditional Accuracy (%)        | 85.5    | 83.9    | 92.3   | 78.1   | 100.0    |
>
>
>
>
> Even with step-by-step reasoning, the model struggles due to missing representations of death and time. **Our prompt performs better by explicitly providing the death year and instructing the model to compare it with the event year.**
>
>
>
> ### **Concept incongruence in an agentic setting**
>
> We agree that concept incongruence plays an important role in agentic systems, and we appreciate your perspective on its broader implications. While the agentic behavior is beyond our scope of study currently, we see this as a valuable direction for future work.

---

### Official Review · Reviewer_ZUxs · 2025-06-30

**Clarity:** 3
**Significance:** 2
**Originality:** 3
**Rating:** 4
**Confidence:** 3

**Summary:**

This paper introduces the concept of "concept incongruence" in large language models (LLMs), where conflicting or incompatible conceptual boundaries—whether between user instructions and internal model representations, or within internal model states—lead to unexpected or inconsistent behaviors. The authors focus on a controlled scenario involving role-play with deceased historical figures and evaluate model responses to time-sensitive queries across “before” and “after death” boundaries. They define three behavioral metrics—abstention rate, answer rate, and conditional accuracy—and use them to analyze model behavior. Despite valuable insights into behavior under temporal-role incongruence (a subset of I-B/I-C incongruence), the study falls short of generalizing its findings to broader, real-world use cases.

**Questions:**

1) Could the behavioral metrics or detection methods generalize to complex, multi-turn dialogue settings where incongruence is implicit or evolves over time?
2) Why wasn’t the I-C level of incongruence studied directly with synthetic or adversarial prompts? Since I-C is arguably the most concerning form from an alignment and safety standpoint, more rigorous attention here would strengthen the contribution.

**Ethical Concerns:**

["NO or VERY MINOR ethics concerns only"]

**Final Justification:**

I'd like to thank the authors for the response which have addressed most of my concerns. I have accordingly adjusted the score.

**Limitations:**

yes

**Quality:**

2

**Strengths And Weaknesses:**

Strengths
1) The paper introduces a new, well-categorized conceptual framework (I-A, I-B, I-C) that articulates how incongruence can arise in LLM behavior. This framing has potential implications for both safety and alignment research.
2) The use of abstention rate, answer rate, and conditional accuracy provides a structured lens for quantifying LLM behavior under constraint violation, and helps to concretely measure effects of incongruence.
3)  The authors use a carefully designed experimental setup with historical figures and time-sensitive queries, enabling clean isolation of effects. Automatic evaluation with GPT-4o-mini and validation by human annotators further adds credibility.


Weaknesses
1) While the controlled setting is clean, the study’s focus is narrow—almost exclusively on a single type of incongruence (temporal inconsistency post-death in role-play). It does not test other real-world concept conflicts, such as task-specific logic (e.g., “make stable prices in a free market”), social role conflicts, or implicit misalignment.
2) The paper defines I-C (conflict within model representations) as the most subtle and challenging form of incongruence, yet it does not directly explore or isolate I-C cases outside the role-play context. This limits the contribution's depth on this potentially most critical front.
3) The study is largely observational. While insightful, it doesn’t propose new learning, alignment, or inference-time methods for mitigating concept incongruence. The suggested mitigation (explicit prompting) is simple and comes with a clear trade-off (accuracy loss), limiting practical utility.

---

> ### Author Rebuttal · Authors · 2025-07-30
>
> Thank you for finding our *concept incongruence* framework novel and well organized. We are glad that you also appreciate the benchmark we created, which centered on time and death in role playing, and recognize the strength of our evaluation metrics, and the clean experiment setup! We address your concerns and questions below.
>
> ### **New experiments on higher-level concept incongruence**
>
> Inspired by your suggestions, we conducted new experiments on CommonsenseQA dataset [1] to demonstrate broader implications of role-play warping model representations of world knowledge. We observe a performance drop under role-play, indicating the conflict between role-play representation and the commonsense knowledge representation, e.g., >10% in Gemma.
>
> |                             |  Llama  |  Gemma  | Claude |  GPT   |
> |:----------------------------|:-------:|:-------:|:------:|:------:|
> | Non-Role-Play Accuracy (%) | 76.7    | 80.0    | 86.7   | 80.0   |
> | Role-Play Accuracy (%)     | 67.6    | 66.8    | 81.2   | 75.8   |
>
> **Together with our results on artworks in paper, this demonstrates that our findings that role-play warping world knowledge with presidential questions have broad implications.**
>
>
>
> ### **Addressing concerns about narrow instantiation**
>
> We focus on temporal boundaries in role-play for two reasons. First, temporal representation provides a measurable instantiation of concept incongruence and enables us to isolate and evaluate representational conflicts that would be harder to detect in more abstract domains such as social role conflicts and implicit misalignment. Second, the role-play context offers a clear and controllable knowledge boundary via known death years and presidential terms, unlike vague boundaries like space and educational background. The role-play context allows for precise evaluation of temporal inconsistency. We view this as a well-scoped starting point and encourage future work to extend to more complex and varied forms of concept incongruence.
>
> **Furthermore, our task design allows us to engage with all three levels within a role-play context.** For example, asking a historical figure about post-death events shows an **I-A** conflict. Probing the model’s internal states reveals a mismatch between human and model representations of death **(I-B)**. Moreover, the observed shifts in the model’s internal timeline during role-play suggest internal representational conflicts **(I-C)**, as the model appears to reconcile world knowledge with the activated persona. In comparison, existing benchmarks like FalseQA [2] mostly capture I-A level conflicts and do not address the deeper representational issues at I-B and I-C.
>
>
>
> ### **Addressing the request on isolating I-C and focusing more on safety and alignment**
>
> We agree that concept incongruence has important implications for safety and alignment, as illustrated in Figure 1. As discussed in the previous response, our task engages all three levels of incongruence in a controlled role-play setting and is intentionally chosen to showcase the relevance and importance of concept incongruence, providing an example for future work to further explore this novel concept.
>
> **While we do not isolate I-C conflicts outside role-play, our findings provide important evidence of internal representational conflict.** This is echoed in alignment faking studies [3], where later-injected goals override earlier ones, and in jailbreaks, where models switch between “harmless” and “helpful” modes depending on which representation dominates. The fact that long system prompts can be overridden by a few user lines [4] further illustrates how concept incongruence manifests in AI safety challenges.
>
>
>
> ### **Addressing the concerns about mitigating concept incongruence**
>
> Our prompting intervention improves abstention and answering behavior, likely because the model becomes more immersed in the role. This immersion shifts its internal timeline toward the character’s context, making the observed accuracy drop expected as world knowledge gives way to role fidelity.
>
> More broadly, our contribution is conceptual. We introduce concept incongruence as a framework for understanding representational conflict in human-LLM interaction. As discussed in Section 7, this phenomenon exposes deeper issues of mis-specification that vary by task and user intent. We think this is not a problem that begs for a quick solution. In some settings, incongruence should be suppressed; in others, such as creative generation, it may be desirable. Rather than falling into solutionism, we advocate for models that can detect internal conflicts and expose them to users, enabling flexible, context-sensitive behavior.
>
>
>
> ### **Clarifying the generalizability of the behavior metrics to multi-turn dialogue settings.**
>
> Different types of concept incongruence may manifest differently, but our behavioral metrics are designed to evaluate cases involving knowledge boundaries. While our current setting focuses on single-turn prompts, our behavioral metrics are generalizable to multi-turn dialogue.
>
>
> [1] Talmor, Alon, et al. "Commonsenseqa: A question answering challenge targeting commonsense knowledge." arXiv preprint arXiv:1811.00937 (2018).
>
> [2] Hu, Shengding, et al. "Won't get fooled again: Answering questions with false premises." arXiv preprint arXiv:2307.02394 (2023).
>
> [3] Ryan Greenblatt, Carson Denison, Benjamin Wright, Fabien Roger, Monte MacDiarmid, Sam383
> Marks, Johannes Treutlein, Tim Belonax, Jack Chen, David Duvenaud, Akbir Khan, Julian384
> Michael, Sören Mindermann, Ethan Perez, Linda Petrini, Jonathan Uesato, Jared Kaplan, Buck385
> Shlegeris, Samuel R. Bowman, and Evan Hubinger. Alignment faking in large language models,386
> 2024.
>
> [4] Zeng, Yi, et al. "How johnny can persuade llms to jailbreak them: Rethinking persuasion to challenge ai safety by humanizing llms." Proceedings of the 62nd Annual Meeting of the Association for Computational Linguistics (Volume 1: Long Papers). 2024.

---

> > ### Comment · Reviewer_ZUxs · 2025-08-06
> >
> > I'd like to thank the authors for the response which have addressed most of my concerns. I have accordingly adjusted the score.

---

> > > ### Author Response · Authors · 2025-08-06
> > > **Thank you!**
> > >
> > > Thank you for acknowledging our response and adjusting your score. We appreciate your thoughtful feedback and are glad we could address most of your concerns. Thanks again for the time and effort you put into reviewing our submission.

---

### Official Review · Reviewer_k8Bq · 2025-07-02

**Clarity:** 3
**Significance:** 3
**Originality:** 3
**Rating:** 3
**Confidence:** 5

**Summary:**

This paper introduces the concept incongruence, i.e., a phenomenon where conflicting concept boundaries in user prompts or model representations lead to underspecified or mis-specified LLM behaviors. Focusing on temporal clashes in role-playing scenarios (e.g., querying dead characters about post-death events), the authors define three metrics (abstention rate, conditional accuracy, and answer rate) to quantify model responses.
This paper finds that (1) LLMs fail to abstain appropriately after a character’s death (abstention rates deviate 81.3–90.4% from expectations). (2)LLMs exhibit gradual behavioral shifts near death years instead of sharp boundaries. (3) LLMs suffer significant accuracy drops (up to 9.2%) compared to non-role-play settings. Probing experiments trace these issues to unreliable encoding of death states and temporal representation shifts during role-playing. While explicit specifications (e.g., adding death-year checks) improve abstention, they further reduce accuracy, highlighting an irreconcilable tension between role-consistency and factual knowledge. The work establishes the first taxonomy of concept incongruence (I-A/B/C), exposes critical LLM vulnerabilities, and urges research into robust conflict resolution.

**Questions:**

1. It is recommended to add more dimensions of knowledge boundaries, in addition to time, there may space, culture, educational background, etc., to explore this issue in depth. It is also recommended to increase the problems and fields of experimental settings.
2. It is necessary to supplement a more in-depth analysis of the impact of internal representation on the abandonment rate, conditional accuracy, and answer rate in role-playing, and then provide a relatively ideal optimization solution.

**Ethical Concerns:**

["NO or VERY MINOR ethics concerns only"]

**Final Justification:**

I had several rounds of discussion with the author, but the author did not dispel my concerns, so I decided to keep my score.

**Limitations:**

yes

**Quality:**

2

**Strengths And Weaknesses:**

Strengths
1. This paper proposes, for the first time, a three-level classification framework for conceptual conflicts, offering a new perspective on studying knowledge boundary issues in role-playing.
2. The paper proposes multi-dimensional evaluation metrics, including Abstention Rate, Conditional Accuracy, and Answer Rate, to reveal conceptual conflicts. It further uses representation probe technologies to locate the missing death state encoding and time representation offset as the root causes.

Weaknesses
1. The experimental data are very limited. Only two questions about the inauguration time of the US president are used for the experiment. Although the artwork data is supplemented, the domain and questions are still very single, resulting in the conclusions not being universal enough.
2. Lack of in-depth analysis of the time representation shift caused by role-playing.
3. The optimized prompt can improve the abstention, but it will reduce the conditional accuracy. The author explains that "The lack of death-year representation, in turn, drives the suboptimal mix of abstention and answer behavior." This part needs to be explained in detail. Since the death year is already specified in the prompt, there must be a relevant time representation, so why does it lead to a decrease in accuracy?

---

> ### Author Rebuttal · Authors · 2025-07-30
>
> Thank you for appreciating our novel *concept incongruence* framework and recognizing the strength of our multi-dimensional evaluation metrics! We address your concerns and questions below.
>
>
> ### **New experiments on broader implications of concept incongruence**
>
> Inspired by your suggestions, we conducted new experiments on CommonsenseQA dataset [1] to demonstrate broader implications of role-play warping model representations of world knowledge. We observe a performance drop under role-play, indicating the conflict between role-play representation and the commonsense knowledge representation beyond time, e.g., >10% drop in Gemma.
>
> |                             |  Llama  |  Gemma  | Claude |  GPT   |
> |:----------------------------|:-------:|:-------:|:------:|:------:|
> | Non-Role-Play Accuracy (%) | 76.7    | 80.0    | 86.7   | 80.0   |
> | Role-Play Accuracy (%)     | 67.6    | 66.8    | 81.2   | 75.8   |
>
> **Together with our results on artworks in paper, this demonstrates that our findings that role-play warping world knowledge with presidential questions have broad implications.**
>
>
>
> ### **Addressing concerns about experiment data and dimension of knowledge boundary**
>
> We focus on temporal boundaries in role-play for two reasons. First, temporal representation provides a measurable instantiation of concept incongruence and enables us to isolate and evaluate representational conflicts that would be harder to detect in more abstract domains. Second, the role-play context offers a clear and controllable knowledge boundary via known death years and presidential terms, unlike vague boundaries like space, cultural and educational background. The role-play context allows for precise evaluation of temporal inconsistency. We view this as a well-scoped starting point and encourage future work to extend to more complex and varied forms of concept incongruence.
>
> **Furthermore, our task design allows us to engage with all three levels within a role-play context.** For example, asking a historical figure about post-death events shows an **I-A** conflict. Probing the model’s internal states reveals a mismatch between human and model representations of death **(I-B)**. Moreover, the observed shifts in the model’s internal timeline during role-play suggest internal representational conflicts **(I-C)**, as the model appears to reconcile world knowledge with the activated persona. In comparison, existing benchmarks like FalseQA [2] mostly capture I-A level conflicts and do not address the deeper representational issues at I-B and I-C.
>
>
>
> ### **Clarifying in-depth analysis of internal temporal and death representations**
>
> We appreciate the opportunity to clarify our analysis. In Section 4, we examine how internal representations contribute to behavioral inconsistencies in detail. Section 4.1 shows that the absence of death and death-year representations affects abstention and answering behavior. Section 4.2 uses an artwork dataset that is distinct from our presidential questions to demonstrate that role-play shifts the model’s temporal representations, leading to greater deviations from the actual timeline.
>
> Further, in Table 2 of the Supplementary Material, we truncate the timeline into 5-year intervals and observe a significant drop in correlation between predictions and ground truth. This suggests that the **model encodes time coarsely, preserving only broad relative order**. As such, simply adding a death-year reference may not suffice to correct the broader temporal shift caused by role-play. Please refer to Supplementary Sections A and B for details.
>
>
> ### **Clarifying how death year representation relates to model accuracy**
>
> As discussed in Section 5 line 279~298, we find that **the lack of an internal death representation primarily affects abstention and answering behavior**. Making the death year explicit improves these behaviors.
>
> However, it does not improve accuracy. As shown in Figure 5b and Supplementary Figure 3, role-play degrades temporal representation. RMSE increases by 8.2 years for Llama and 3.2 years for Gemma. This suggests that stronger role immersion compromises the model’s ability to retain accurate world knowledge, revealing a trade-off between role-play faithfulness and world knowledge (further confirmed by our newly added experiments on CommonsenseQA above). Please refer to Supplementary Section A for more details.
>
> [1] Talmor, Alon, et al. "Commonsenseqa: A question answering challenge targeting commonsense knowledge." arXiv preprint arXiv:1811.00937 (2018).
>
> [2]  Hu, Shengding, et al. "Won't get fooled again: Answering questions with false premises." arXiv preprint arXiv:2307.02394 (2023).

---

> ### Author Response · Authors · 2025-08-07
>
> Dear Reviewer k8Bq,
>
> Thank you very much for your thoughtful comments and insightful suggestions. We sincerely appreciate the time and effort you have dedicated to reviewing our work.
>
> Based on your review, we have included additional experiments and clarifications in our rebuttal. Can you please tell us if our rebuttal addresses your concerns?
>
> If you have any further questions or concerns, we would be happy to provide additional details. We are truly grateful for your time and consideration.
>
> Best,
> The Authors

---

> > ### Comment · Reviewer_k8Bq · 2025-08-08
> > **Response to Authors**
> >
> > I'm very grateful for the authors' additional experiments and clarifications. This paper introduces concept incongruence and specifically explores conceptual boundaries in role-playing scenarios, specifically temporal boundaries. This type of conceptual inconsistency is particularly prominent in role-playing and has been extensively studied. It's well-established that models often struggle to understand temporal boundaries during role-playing. The authors' solution, as a conventional approach, can alleviate this problem to some extent. In this respect, the paper doesn't present any new findings or solutions.
> >
> > Furthermore, concept incongruence in role-playing involves many other boundary issues besides temporal boundaries, such as space, cultural, and cognitive boundaries. Although these boundaries are not clear, using temporal boundaries as an example may be overly restrictive in exploring conceptual incongruence.
> >
> > I also have a question: It's theoretically important for models to establish clear temporal boundaries during role-playing. For example, it seems reasonable that the character shouldn't know about the president after the character's death, given the clear timeline. However, if, for example, the character is playing Henry II and is asked about the Riemann Hypothesis, should the model refuse to answer? On a technical level, it's crucial for models to maintain clear boundaries when playing roles. However, at the application level, how should we define conceptual incongruence? Can a one-size-fits-all approach to temporal boundaries truly resolve conceptual incongruence?
> >
> > I believe that a more comprehensive boundary analysis and solutions would make the paper more valuable.

---

> > > ### Author Response · Authors · 2025-08-08
> > >
> > > Thank you for your response! We address your concerns and questions below.
> > >
> > > ### **How Our Work Differs from Prior Studies**
> > >
> > > Prior studies on role-play only study the performance drop in fact-related questions [1, 2, 3]. None of them dived into the reason why such performance drop happens, partly because of your second point: it is challenging to formulate the problem with many different factors. In contrast, **our work goes beyond accuracy and examines temporal representations in the model**. Our work is the first one to pinpoint the problem of concept incongruence.
> > >
> > > Specifically, we define three levels of concept incongruence, analyze them by constructing a benchmark centered on death in role-playing and real-world time, and examine them through the lens of internal representations.
> > >
> > > To the best of our knowledge, we are the first to formalize the phenomenon within a novel concept incongruence framework, investigate the underlying internal representation conflicts, and discuss the extension of this behavior to other contexts. **Could you point us to specific citations and specify what you mean by extensive studies?**
> > >
> > > ### **Why We Focus on Temporal Boundaries**
> > >
> > > As noted in our rebuttal, we focus on temporal boundaries in role-play because **they offer a concrete, measurable case of concept incongruence and make representational conflicts easier to detect than in abstract domains**. Known death years and presidential terms provide clear, documented knowledge boundaries for precise evaluation of temporal inconsistency. In contrast, boundaries such as cognition, space, or culture are harder to define or poorly documented. For example, your question about the Henry II–Riemann Hypothesis scenario might be about the difficulty of specifying cognitive boundaries. But honestly, there could be multiple incongruences in this example, which again showcases our point, so we do not know what your expected behavior should be. Therefore, our clean problem formulation is crucial for this study.
> > >
> > > ### **Concept Incongruence as a Specification Problem**
> > >
> > > We emphasize that resolving concept incongruence is not the goal, and that is why we wrote in Section 7 that **concept incongruence often stems from mis-specification or underspecification of desired behavior**. Without a clear understanding of the user’s goal, the model may adopt inconsistent strategies for resolving conflicts.
> > >
> > >
> > > In the original review, both you and Reviewer ZUxs requested a solution for mitigating concept incongruence (e.g., you suggested that our paper could benefit from “a relatively ideal optimization solution”). However, as you suggested in your new response, there is unlikely to be a “one-size-fits-all” approach to resolving such conflicts. **We agree with your updated position.**
> > >
> > > As we stated in our rebuttal to Reviewer ZUxs (point 4), **the presence of concept incongruence can also be informative and valuable in creative domains**. **Mitigation is therefore not the goal, and our paper does not aim for resolution.** Instead, we envision models that can detect concept incongruence and, with user guidance, decide whether to resolve them in factual or safety-critical contexts or preserve them for open-ended, imaginative uses.
> > >
> > > ---
> > >
> > > [1] Zhang, Wenyuan, et al. "Revealing and Mitigating the Challenge of Detecting Character Knowledge Errors in LLM Role-Playing." arXiv preprint arXiv:2409.11726 (2024).
> > >
> > > [2] Gupta, Shashank, et al. "Bias runs deep: Implicit reasoning biases in persona-assigned llms." arXiv preprint arXiv:2311.04892 (2023).
> > >
> > > [3] Zheng, Mingqian, et al. "When” a helpful assistant” is not really helpful: Personas in system prompts do not improve performances of large language models." Findings of the Association for Computational Linguistics: EMNLP 2024. 2024.

---

> > > > ### Comment · Reviewer_k8Bq · 2025-08-09
> > > > **Response to Authors**
> > > >
> > > > I agree with the statement that the presence of concept incongruence can also be informative and valuable in creative domains.
> > > >
> > > > The authors state that this paper aims to detect concept incongruence, but temporal boundaries are not the same as concept incongruence. Factual or safety-critical contexts do not equate to strictly controlling temporal boundaries, nor do imaginative uses equate to softening temporal boundaries. The authors' first study of concept incongruence and their definition of three levels of concept incongruence are indeed innovative. The paper also explores and analyzes temporal boundaries in detail, a feat previously unseen. However, these findings are not new. The authors' proposed detection benchmark is also specific to temporal boundaries and difficult to extend to other boundaries. So, is the authors' claim that this paper can detect concept incongruence a bit exaggerated?

---

> > > > > ### Author Response · Authors · 2025-08-09
> > > > >
> > > > > Thanks for replying again!
> > > > >
> > > > > We never claimed that we have a method for detecting general concept incongruence. Our claim is that we define three levels of concept incongruence, analyze them by constructing a benchmark centered on death in role-playing and real-world time, and examine them through the lens of internal representations.
> > > > >
> > > > > Also, you have been arguing that our findings are not new. Could you point us to specific references with specific findings to support your argument?

---

### Official Review · Reviewer_u4PE · 2025-07-02

**Clarity:** 3
**Significance:** 3
**Originality:** 3
**Rating:** 5
**Confidence:** 4

**Summary:**

This paper studies how language models behave when the concepts involved in a prompt are in conflict, focusing on role-play prompts where the model is asked for information while assuming the role of someone who could not have known that information. They observe accuracy drops in this setting, and argue that this arises from a clash between “character immersion” and world knowledge, finding evidence in the model’s hidden representations.

**Questions:**

- nit: I feel the opening example in the abstract should be about the core experiment (e.g. asking Marilyn Monroe about US presidents after her death), rather than about the unicorn with two horns, which is never explored.
- I think the authors should consider citing “Bias runs deep: Implicit reasoning biases in persona-aligned llms” (Gupta 2023) - it shows that assuming different roles impacts a model’s capabilities.
- 4.1 - Authors acknowledge that the expected behavior is not clear in their primary setting (ie maybe the prompt is asking for Marilyn Monroe to hallucinate a response, or maybe it’s asking for the model to mimic other aspects of Marilyn Monroe, but still provide the answer). I think this is also true for the question “Are you dead or alive?” - maybe the desired behavior is that the model thinks the character is alive when role playing.
- 4.1 - I think more details would be nice on the probing experiments. What was the dead vs alive breakdown of characters in the training set for (i)?
- 4.2 - the artwork example highlights the problem where the expected behavior is unclear - e.g. maybe it makes sense that Marilyn Monroe doesn’t know the years when artworks were released. It’s interesting that the accuracy drop for artwork questions is greater than for questions about US presidents - suggests that “getting into character” restricts world knowledge to varying degrees.
- 5 - this result makes me wonder whether chain of thought would improve behavior. I also wonder whether the condition studied, minus the part where the death year is explicitly provided, would provide the same gains, as section 4 shows models know the death year.
- 7 - The claim that “models lack a robust internal representation of death” I feel is overstated and underspecified.
- nit: section 3 - which model is responsible for the Marie Curie example?
- nit: fig 2 - maybe color and pattern would be helpful for encoding different models
- nit: fig 2 maybe accuracy should instead be difference in accuracy - I realize baseline accuracy is 100% across the board but it seems like it’s the difference that should be highlighted
- nit: fig 3 - there does to me seem to be a sharp shift around index 0. I also wonder whether there would be a directional trend if you studied older figures (with greater than 30 positive indices)?
- nit: fig 4 - a fuller caption would be nice - explaining how accuracy is computed.

**Ethical Concerns:**

["NO or VERY MINOR ethics concerns only"]

**Final Justification:**

I stand by my rating of 5: Accept: Technically solid paper, with high impact on at least one sub-area of AI or moderate-to-high impact on more than one area of AI, with good-to-excellent evaluation, resources, reproducibility, and no unaddressed ethical considerations.

The author's rebuttal was comprehensive, and addressed many of my concerns. I appreciate that the authors presented many new results, e.g. about how adopting personas in role play hurts CommonSense QA performance. However I still feel that the experimental setting is somewhat narrow, given existing literature on the effects of role-playing, the primary observation about decreased accuracy is expected.

**Limitations:**

yes

**Quality:**

3

**Strengths And Weaknesses:**

Strengths:
- How models behave under “concept incongruence” is a significant and understudied area, and the authors offer a framework that operationalizes the problem.

Weaknesses:
- While the problem of concept incongruence is interesting and significant, the authors study a very narrow instantiation. They acknowledge this under “Limitations”.
- The paper presents many interesting results - however the core thread was provided by the narrow experimental setting of asking about US presidents, rather than the more interesting high-level question of concept incongruence. I feel like the paper taught me a lot about how models behave when role-playing historical figures and answering questions about US presidents, but I wanted more discussion and proof of how this experimental setting is more broadly instructive. Maybe the issue is that the experiment delves into both concept incongruence and temporal representations, two very rich areas, and as a result there was less room for making the broader case.

---

> ### Author Rebuttal · Authors · 2025-07-30
>
> Thank you very much for appreciating our framework for *concept incongruence* and interesting findings that current models do not demonstrate desirable abstention behavior and present a drop in accuracy when role playing. We address your concerns and questions below.
>
> ### **Concerns about narrow instantiation**
>
> We focus on temporal boundaries in role-play for two reasons. First, temporal representation provides a measurable instantiation of concept incongruence and enables us to isolate and evaluate representational conflicts that would be harder to detect in more abstract domains. Second, the role-play context offers a clear and controllable knowledge boundary via known death years and presidential terms, allowing for precise evaluation of temporal inconsistency. We view this as a well-scoped starting point and encourage future work to extend to more complex and varied forms of concept incongruence.
>
> **Furthermore, our task design allows us to engage with all three levels within a role-play context.** For example, asking a historical figure about post-death events shows an **I-A** conflict. Probing the model’s internal states reveals a mismatch between human and model representations of death **(I-B)**. Moreover, the observed shifts in the model’s internal timeline during role-play suggest internal representational conflicts **(I-C)**, as the model appears to reconcile world knowledge with the activated persona. In comparison, existing benchmarks like FalseQA [1] mostly capture I-A level conflicts and do not address the deeper representational issues at I-B and I-C.
>
>
> ### **New experiments on higher-level concept incongruence**
>
> Inspired by your suggestions, we conducted new experiments on CommonsenseQA dataset [2] to demonstrate broader implications of role-play warping model representations of world knowledge. We observe a performance drop under role-play, indicating the conflict between role-play representation and the commonsense knowledge representation beyond time, e.g., >10% drop in Gemma.
> |                             |  Llama  |  Gemma  | Claude |  GPT   |
> |:----------------------------|:-------:|:-------:|:------:|:------:|
> | Non-Role-Play Accuracy (%) | 76.7    | 80.0    | 86.7   | 80.0   |
> | Role-Play Accuracy (%)     | 67.6    | 66.8    | 81.2   | 75.8   |
>
>
>
> **Together with our results on artworks in paper, this demonstrates that our findings that role-play warping world knowledge have broad implications.**
>
> These findings also echo prior research on how personas affect model performance. Gupta et al. [3] shows that adopting personas from different demographic groups can negatively impact reasoning task accuracy. Similarly, Zheng et al. [4] demonstrated that persona characteristics can significantly influence prediction accuracy. While these studies investigate model performance under role-play, our work formalizes the phenomenon within the framework of concept incongruence and examines the temporal representations beyond the accuracy.
>
>
>
> ### **Implementing additional reasoning techniques and prompting format**
>
> Inspired by your suggestions, we have implemented chain-of-thought reasoning by using a standard prompt with an extra instruction to ask the model to "think step by step". **While CoT slightly improves accuracy, its behavior is worse in abstention and answer rate compared to our designed prompt.**
>
> Result of standard prompt (in the paper):
> |                                  | Llama | Gemma | Claude | GPT  | Expected |
> |:----------------------------------|:-------:|:--------:|:--------:|:------:|:----------:|
> | Abstention (Before) (%)         |  4.6  |  0.0   |  0.7   | 2.6  | 0.0      |
> | Abstention (After) (%)          | 18.7  |  1.9   |  9.6   | 0.4  | 100.0      |
> | Answer (Before) (%)             | 99.3  | 100    | 99.3   | 97.5 | 100.0      |
> | Answer (After) (%)              | 93.8  | 98.1   | 92.6   | 99.6 | 0.0      |
> | Conditional Accuracy (%)        | 92.0  | 92.0   | 99.6   | 91.8 | 100.0      |
>
>
>
> Results of standard prompt + CoT:
> |                                  |  Llama  |  Gemma  | Claude |  GPT   | Expected |
> |:--------------------------------|:-------:|:-------:|:------:|:------:|:--------:|
> | Abstention (Before) (%)         |  9.2    |  43.6   | 32.3   |  2.2   |   0.0    |
> | Abstention (After) (%)          | 21.8    |  54.9   | 57.8   |  0.6   | 100.0    |
> | Answer (Before) (%)             | 99.4    |  72.8   | 98.4   | 99.7   | 100.0    |
> | Answer (After) (%)              | 94.7    |  64.6   | 89.0   | 99.4   |   0.0    |
> | Conditional Accuracy (%)        | 93.5    |  93.8   | 96.8   | 98.2   | 100.0    |
>
>
>
> Results of our designed restricted prompt (in the paper):
> |                                  |  Llama  |  Gemma  | Claude |  GPT   | Expected |
> |:--------------------------------|:-------:|:-------:|:------:|:------:|:--------:|
> | Abstention (Before) (%)         |  4.8    |  0.0    |  1.6   |  0.0   |   0.0    |
> | Abstention (After) (%)          | 94.2    | 68.0    | 98.0   | 87.8   | 100.0    |
> | Answer (Before) (%)             | 97.8    | 100.0   | 99.8   | 91.1   | 100.0    |
> | Answer (After) (%)              | 10.2    | 32.0    | 17.6   | 24.3   |   0.0    |
> | Conditional Accuracy (%)        | 85.5    | 83.9    | 92.3   | 78.1   | 100.0    |
>
>
> As suggested, we have also used the restricted prompt we studied, but without the death year provided, as a comparison. The results show that it is less effective than our designed prompt. We’d also like to clarify that in Section 4, we find that the model does not have an accurate death (line 207) or death year representation (line 217). With these insights, we designed the prompt that introduces the death year explicitly.
>
> |                                  |  Llama  |  Gemma  | Claude |  GPT   | Expected |
> |:--------------------------------|:-------:|:-------:|:------:|:------:|:--------:|
> | Abstention (Before) (%)         |  6.5    |  0.5    | 13.6   | 15.3   |   0.0    |
> | Abstention (After) (%)          | 89.1    | 59.7    | 98.6   | 63.3   | 100.0    |
> | Answer (Before) (%)             | 95.7    | 99.5    | 96.8   | 85.5   | 100.0    |
> | Answer (After) (%)              | 67.3    | 40.3    | 32.6   | 41.5   |   0.0    |
> | Conditional Accuracy (%)        | 93.7    | 93.7    | 99.6   | 77.3   | 100.0    |
>
> Even with step-by-step reasoning, the model struggles due to missing representations of death and time. **Our prompt performs better by explicitly providing the death year and instructing the model to compare it with the event year.**
>
>
>
> ### **Clarifying expected behavior in some experiments**
>
> The prompt “Are you dead or alive?” was designed to surface behavioral incongruence. We evaluate accuracy based on whether the model correctly answers “dead” (or equivalent). For artwork questions, we expect correct answers before the death year and abstention after. To simplify evaluation, we do not account for educational or cultural background.
>
> Our results show that models lack a consistent strategy, sometimes answering what the character shouldn’t know, other times abstaining inconsistently. This adds to the evidence that the model lacks accurate representations of death, leading to divergent behavior across similar prompts.
>
>
>
> ### **Clarifying probing details**
>
> We introduce the detailed setup of probing in the Supplementary Materials. For the layer-wise “dead / alive” probe, we compile 1,000 dead and 1,000 alive individuals. We split 80%/20% for training and testing.  We train our linear probes on the final-token hidden states across layers. The accuracy we measured in Figure 4 is the test accuracy of the probe. Please refer to Supplementary Materials Section A for more details.
>
>
>
> ### **Addressing request on extending the time range**
>
> We selected 100 historical figures who died between 1890 and 1993 to test model behavior 30 years before and after death. Expanding this range would reduce dataset coverage. While behavior improves further from the death year, abstention tops out near 50% and answer rate remains around 60%, both far from ideal (abstention:100%, answer: 0%).
>
>
>
> ### **Writing suggestions**
>
> We are grateful for the suggestions and will incorporate them in the revision.
>
>
> [1]  Hu, Shengding, et al. "Won't get fooled again: Answering questions with false premises." arXiv preprint arXiv:2307.02394 (2023).
>
> [2] Talmor, Alon, et al. "Commonsenseqa: A question answering challenge targeting commonsense knowledge." arXiv preprint arXiv:1811.00937 (2018).
>
> [3] Gupta, Shashank, et al. "Bias runs deep: Implicit reasoning biases in persona-assigned llms." arXiv preprint arXiv:2311.04892 (2023).
>
> [4] Zheng, Mingqian, et al. "When” a helpful assistant” is not really helpful: Personas in system prompts do not improve performances of large language models." Findings of the Association for Computational Linguistics: EMNLP 2024. 2024.

---

### Author Response · Authors · 2025-08-04
**We would love to discuss!**

Dear reviewers,

Thank you again for your thoughtful reviews. With only a few days remaining in the discussion period, we would greatly appreciate it if you could kindly acknowledge our rebuttal. We’d love to discuss any further questions.

Thank you for your time!

---

### Note · Authors · 2025-08-11

Dear reviewers and ACs,

Thank you very much for reviewing and handling our submission! We appreciate that all reviewers recognized **our novel framework for concept incongruence**, **the strength of our multi-dimensional evaluation**, and **the clarity of our analysis**.

During rebuttal, we added experiments based on reviewer suggestions:
* To address the concern about higher-level concept incongruence, we extend our work from temporal boundary to commonsense knowledge. We observe a significant drop in accuracy. Together with our results in the paper, this demonstrates that our findings that role-play warping world knowledge have broad implications.
* Following the suggestions on reasoning techniques, we implemented our experiments with chain-of-thought (CoT). While CoT slightly improves accuracy, its behavior is worse in abstention and answer rate compared to our designed prompt.
* To address the concern about multiple rollouts, we conducted the experiments at `temperature=0.6` with 3 rollouts. The results are very close to our original results, which indicates that this choice is not critical for our experiments

During the discussion period, we addressed most concerns. Although time constraints prevented completing the discussion with Reviewer k8Bq, we believe the discussion helped clarify points from the original review.

In general, we view our work as the first to pinpoint the problem of concept incongruence, not to propose mitigation. We’d like to summarize our contributions as follows:
* We introduce concept incongruence and provide the first systematic categorization to illustrate the space of problems.
* We create a benchmark centered on time and death in role playing and show that current models do not demonstrate desirable abstention behavior and present a drop in accuracy when role playing.
* We find that the inconsistent behavior emerges due to the lack of reliable representations of death and the clash between role playing and world knowledge in the model’s internal representations.

Thanks again for the time and effort you put into reviewing our submission!

---

### Decision · Program_Chairs · 2025-09-17

**Decision:**

Accept (poster)

**Comment:**

This paper introduces and formalizes "concept incongruence" in LLMs and analyzes how conflicting concepts in prompts can cause unexpected behavior by focusing on a temporal role-playing scenario. The reviews were largely positive, with a majority of reviewers recommending acceptance. The primary initial concern, shared by several reviewers (u4PE, k8Bq, ZUxs), was the narrow experimental scope. The authors' rebuttal additional experiments on the CommonsenseQA dataset addressed this concern by demonstrating broader implications and justified the focused setting as a carefully-scoped foundational study. This convinced reviewers ZUxs and HBwf to raise their scores. Although one reviewer (k8Bq) remained unconvinced after a lengthy discussion, the overall consensus is that the paper introduces a novel and important framework. I agree with the reviewers and believe that this paper would be a valuable contribution to NeurIPS. I recommend acceptance.